# Cooperative unfolding of distinctive mechanoreceptor domains transduces force into signals

Lining Ju[1,2,3,4†], Yunfeng Chen[2,5†], Lingzhou Xue[6], Xiaoping Du[7], Cheng Zhu[1,2,5*]

[1]Coulter Department of Biomedical Engineering, Georgia Institute of Technology, Atlanta, United States; [2]Petit Institute for Bioengineering and Biosciences, Georgia Institute of Technology, Atlanta, United States; [3]Heart Research Institute, Camperdown, Australia; [4]Charles Perkins Centre, The University of Sydney, Camperdown, Australia; [5]Woodruff School of Mechanical Engineering, Georgia Institute of Technology, Atlanta, United States; [6]Department of Statistics, The Pennsylvania State University, University Park, United States; [7]Department of Pharmacology, College of Medicine, University of Illinois at Chicago, Chicago, United States

*For correspondence: cheng. zhu@bme.gatech.edu

†These authors contributed equally to this work

Competing interests: The authors declare that no competing interests exist.

**Abstract** How cells sense their mechanical environment and transduce forces into biochemical signals is a crucial yet unresolved question in mechanobiology. Platelets use receptor glycoprotein Ib (GPIb), specifically its $\alpha$ subunit (GPIb$\alpha$), to signal as they tether and translocate on von Willebrand factor (VWF) of injured arterial surfaces against blood flow. Force elicits catch bonds to slow VWF–GPIb$\alpha$ dissociation and unfolds the GPIb$\alpha$ leucine-rich repeat domain (LRRD) and juxtamembrane mechanosensitive domain (MSD). How these mechanical processes trigger biochemical signals remains unknown. Here we analyze these extracellular events and the resulting intracellular $Ca^{2+}$ on a single platelet in real time, revealing that LRRD unfolding intensifies $Ca^{2+}$ signal whereas MSD unfolding affects the type of $Ca^{2+}$ signal. Therefore, LRRD and MSD are analog and digital force transducers, respectively. The >30 nm macroglycopeptide separating the two domains transmits force on the VWF–GPIb$\alpha$ bond (whose lifetime is prolonged by LRRD unfolding) to the MSD to enhance its unfolding, resulting in unfolding cooperativity at an optimal force. These elements may provide design principles for a generic mechanosensory protein machine.

## Introduction

Platelets can serve as a natural model system for studying cell mechanosensing as they rapidly respond to changes in hydrodynamic forces and substrate stiffness due to vascular pathology (*Jackson, 2011*; *Qiu et al., 2015*). Previous studies have suggested the role of GPIb$\alpha$ as a mechanoreceptor, for force exerted on it via its ligand VWF induces platelet signaling (*Ruggeri, 2015*). Conceptually, this coupled mechanical-biochemical process (mechanosensing) can be broken down into four steps: 1) Mechanopresentation: the receptor binding domain A1 is exposed by structural changes in VWF induced by elongational flow and collagen immobilization (*Ju et al., 2015a*; *Springer, 2014*); 2) Mechanoreception: GPIb$\alpha$ LRRD receives the force signal via engaging VWF-A1 to tether the platelet against shear stress; 3) Mechanotransmission: force is propagated from the LRRD through the mucin-like macroglycopeptide (MP) stalk (cf. Figure 2A) (*Fox et al., 1988*) and the MSD across the membrane to adaptor and signaling molecules (e.g. 14-3-3ζ) inside the platelet (cf. Figure 7G); and 4) Mechanotransduction: force induces mechano-chemical changes to

**eLife digest** Platelets – the blood clotting cells – have the ability to detect, interpret and respond to mechanical forces, such as those generated by the flow of blood. The magnitude and duration of the forces detected by the platelets influences whether they form a blood clot. Understanding how the platelets respond to mechanical forces is therefore crucial for our knowledge of conditions such as thrombosis, where blood clots form inside vessels and block them. Clots that form within arteries are associated with heart attack and stroke, which account for around one third of all deaths worldwide.

Cells can sense external forces via individual proteins on their surface and transmit the mechanical information across the cell membrane. This triggers signals within the cell that influence how it responds. However, the molecular details of these "mechanosensory" processes remain poorly understood.

To patch up damaged blood vessels, platelets use a protein on their surface named glycoprotein Ibα (GPIbα) to bind to a plasma protein called von Willebrand factor that adheres to the vessel wall. This binding tethers the platelet to the blood vessel and activates it during clot formation. Previous studies suggested that mechanical force affects how this binding triggers the signals that activate platelets.

Ju, Chen et al. used a homebuilt nanotool to pull on platelet GPIbα while it was bound to von Willebrand factor. This revealed that two distinct domains of the GPIbα protein unfold to relay information about the force, such as its magnitude and duration, to the platelet to trigger biochemical signalling inside the cell. The unfolding of each GPIbα domain has a distinct role in determining the quantity and quality of the signals. The unfolding events work synergistically – they occur together to produce an effect that's greater than the sum of their individual effects. However, pulling on GPIbα via a mutant form of von Willebrand factor eliminated the synergy between the two unfolding events, therefore hindering the effective conversion of mechanical forces into biochemical signals.

Notably, the two GPIbα domains unfolded by force exist in many protein families, including those involved in mediating cell adhesion and detecting signals. The biophysical tools developed by Ju, Chen et al. could be extended to analyze how mechanical cues are presented, received, transmitted and converted into biochemical signals in other cell types and biological systems. Furthermore, the structural insights gained from the platelet GPIbα system may help to design a generic mechanosensory protein machine.

convert mechanical cues to biochemical signals. Some of these steps have been characterized separately. For example, GPIbα forms catch-slip bonds with wild-type (WT) A1 in >15 pN, such that the bond lifetime first increases with force, reaches a maximum at ~25 pN, and decreases thereafter; whereas it forms slip-only bonds with type 2B von Willebrand disease (VWD) mutant (e.g. A1R1450E), such that the bond lifetime decreases monotonically with force (*Ju et al., 2013*; *Yago et al., 2008*). As another example, force induces unfolding of the LRRD, which prolongs A1–GPIbα bond lifetime (*Ju et al., 2015b*), and of the MSD, which is hypothesized to play a role in platelet signaling (*Zhang et al., 2015*). However, how these inter-connected steps are orchestrated to enable the information encoded by force to be translated into biochemical signals is still poorly understood.

We used a biomembrane force probe (BFP) to recapitulate the above process in a single-cell and single-molecular bond level to address the following questions: 1) What molecular events would be induced in GPIbα and how these events are regulated mechanically? 2) Whether, and if so, how changes in presentation of force by VWF-A1 mutation would affect the force reception by GPIbα and its response to force? 3) What features of the force (waveforms) could be sensed by the platelet via GPIbα to initiate intraplatelet calcium fluxes? 4) What proximal events may be responsible for transducing force into a biochemical signal? By manipulating the mechanopresentation and mechanoreception steps then analyzing the resulting mechanotransmission and mechanotransduction steps, we gained insights into the inner workings of this GPIbα-mediated mechanosensory machine.

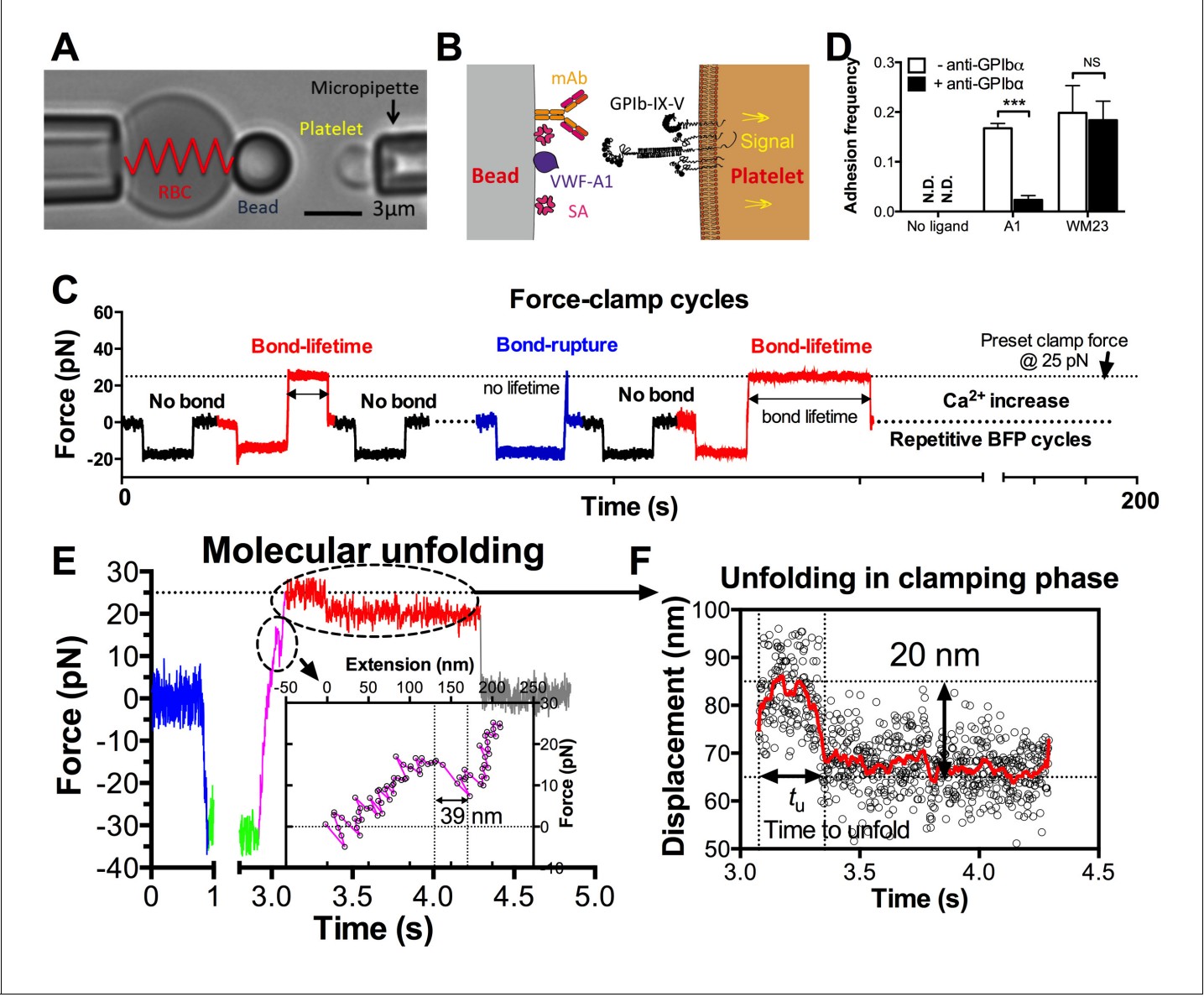

**Figure 1.** BFP analysis of ligand binding kinetics and domain unfolding mechanics of platelet GPIbα. (A) BFP micrograph. A micropipette-aspirated RBC with a probe bead attached to the apex (*left*) was aligned against a platelet held by an opposing micropipette (*right*). (B) BFP functionalization. The probe bead was coated with streptavidin (SA, for attachment to the biotinylated RBC) and VWF-A1 or mAb (*left*) for interaction with platelet GPIb (*right*). (C) Representative force vs. time traces of repetitive force-clamp cycles over a 200-s period. Cycles produced different results are color-coded (black: no bond; blue: bond-rupture; red: bond-lifetime). (D) Mean ± s.e.m. of adhesion frequencies ($n \geq 3$) of platelets binding to beads functionalized with indicated proteins in the absence (open) or presence (closed) of 50 µg/ml AK2. The coating densities are 131 and 95 µm$^{-2}$ for A1 and WM23 respectively. *** = p < 0.001 by Student t-test. (E) Force vs. time trace of a representative BFP cycle showing unfolding signatures in both ramping and clamping phases. The inset zooms in the ramped unfolding signature and indicates the unfolding length. (F) Zoom-in view of the clamped unfolding signature in (E). Higher displacement resolutions were obtained after smoothing the raw data (points) by the Savitzky-Golay method (curves). Time to unfolding ($t_u$) is indicated.

The following figure supplements are available for figure 1:

**Figure supplement 1.** BFP test cycle.

**Figure supplement 2.** Fitting of the WLC model to the force-extension traces (*Figure 1E* insert) before (blue) and after (red) the observed GPIbα ramped unfolding event.

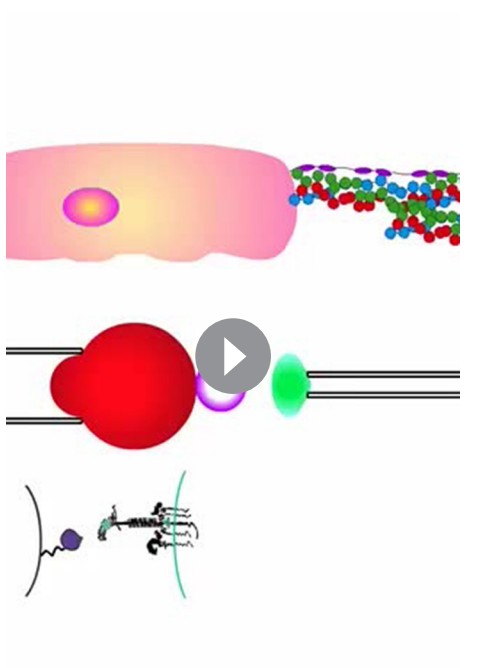

**Video 1.** BFP experiment mimics platelet translocation on sub-endothelium. This animation (produced by Adobe Flash; 12 fps) illustrates the resemblance between platelet translocation on the sub-endothelium (a collagen network covered with VWF on the surface, upper panel) and the repetitive BFP experiment cycle (middle panel) with synchronized molecular interaction between a GPIbα and a VWF-A1 domain (lower panel). Two zoom-in platelet signaling scenarios are inserted following a short- and a long-lived VWF–GPIbα bond respectively. It starts with platelet translocation along the shear force direction, mimicked by the first two no-adhesion BFP cycles. An A1–GPIbα binding event with no lifetime in the BFP cycle results in a transient deceleration in the platelet translocation. After two more no-bond cycles, another bond event survives for a short lifetime without GPIbα unfolding. This triggers a signal (represented by purple stars) displaying a β-type $Ca^{2+}$. Later, another bond event survives for a long lifetime, during which both LRRD and MSD unfold. This triggers a signal (represented by blue stars) displaying an α-type $Ca^{2+}$.

## Results

In the BFP setup, a probe bead was functionalized with VWF-A1 or an anti-GPIb monoclonal antibody (mAb) to serve as a surrogate subendothelial surface (*Figure 1A,B*). It was attached to the apex of a micropipette-aspirated red blood cell (RBC) to form an ultrasensitive force transducer (*Liu et al., 2014*). A platelet was aspired by the target pipette to contact the bead in repetitive force-ramp or force-clamp cycles to mimic the sequential formation, force loading, and dissociation of VWF–GPIbα bonds during the translocation of a platelet on the sub-endothelium (*Video 1*; *Figure 1C* and *Figure 1—figure supplement 1*; Materials and methods). Adhesion frequencies from these cycles were kept low (<20%) by adjusting the ligand or antibody density, a condition required for the platelet to be pulled predominantly (>89%) by a single GPIbα bond (*Chesla et al., 1998*; *Zhu et al., 2002*). Control experiments using beads lacking ligand showed no binding, and blocking with mAb AK2 (epitope mapped to leucine-rich repeat 1–2 overlapping the A1 binding site, cf. *Figure 2A*) eliminated GPIbα binding to A1 but not to mAb WM23 (epitope mapped to the MP below LRRD, cf. *Figure 2A*) (*Figure 1D*) (*Dong et al., 2001*). This confirmed binding specificity and that the binding site of A1 is within LRRD but the binding site of WM23 is outside (*Zhang et al., 2015*).

## Identification of LRRD and MSD unfolding

Using an optical trap, Zhang et al. observed force-induced MSD unfolding in purified recombinant full-length GPIb-IX and a GPIbα stalk region construct (*Zhang et al., 2015*). Using a BFP, we observed LRRD unfolding in glycocalicin (GC) (*Ju et al., 2015b*), the extracellular segment of GPIbα lacking the MSD (*Liang et al., 2013*) (*Figure 2A–C*). Here we pulled GPIbα on platelets via A1 and observed two unfolding signatures, one in the ramping and the other in the clamping phases of the force trace (*Figure 1E*). Unfolding that occurred in the ramping phase is termed ramped unfolding, which is featured by a sudden force kink at 5–20 pN as observed in previous studies of GPIbα unfolding (*Ju et al., 2015b*; *Zhang et al., 2015*). Similar to findings of protein unfolding studies (*Kellermayer et al., 1997*; *Rief et al., 1997*; *Tskhovrebova et al., 1997*; *Zhang et al., 2009a, 2015*), both the force-extension curves before and after unfolding were well fitted by the worm-like chain (WLC) model (*Figure 1—figure supplement 2*). Unfolding that occurred in the clamping phase is termed clamped unfolding, which is featured by an abrupt force drop (*Figure 1F*). Although not observed in the previous studies of GPIbα unfolding (*Ju et al., 2015b*; *Zhang et al., 2015*), this feature has been described in protein unfolding studies using force-clamp experiments (*Oberhauser et al., 2001*; *Tskhovrebova et al., 1997*).

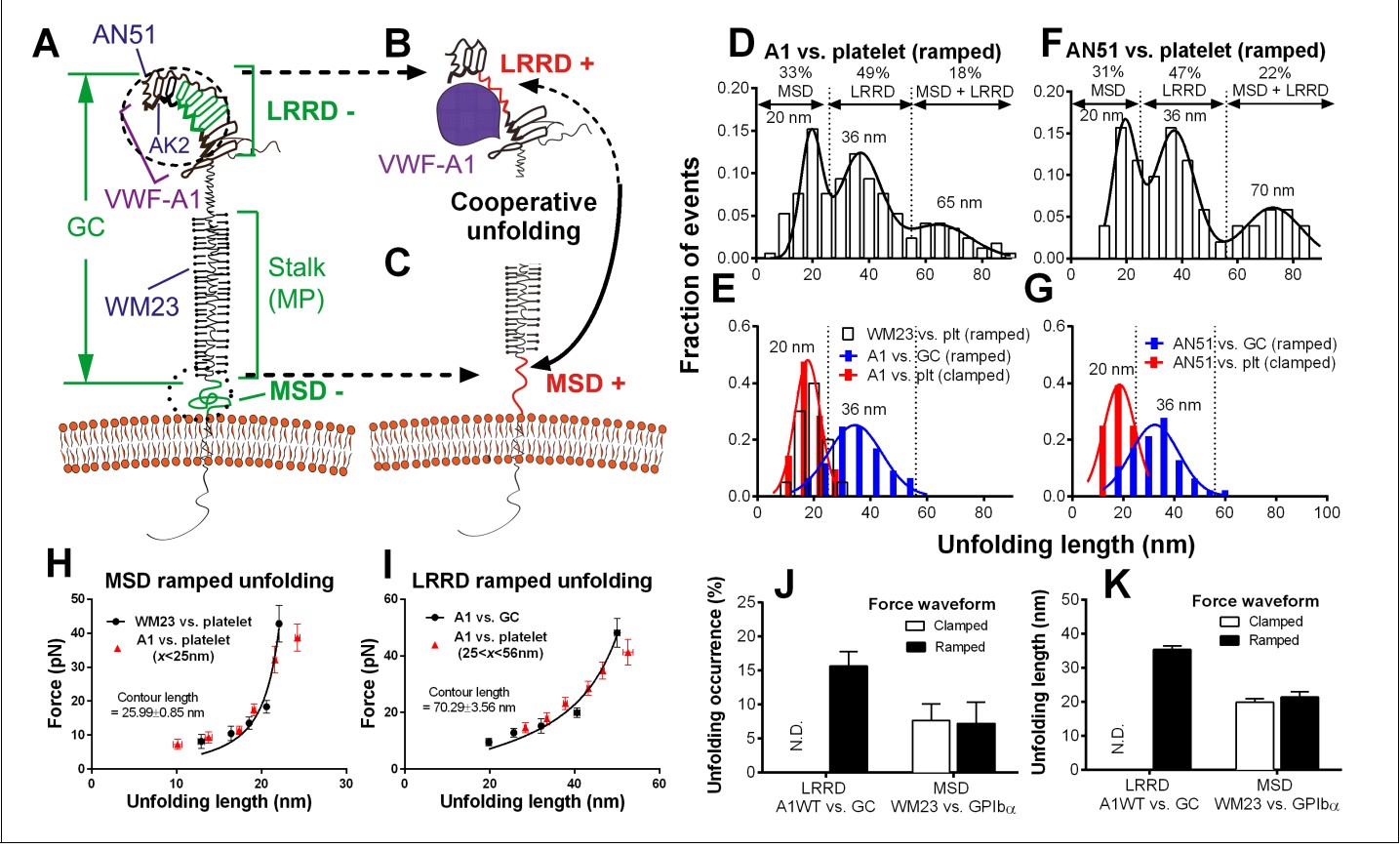

**Figure 2.** Identification and characterization of unfolding of LRRD and MSD. (**A–C**) Schematics of GPIbα on the platelet membrane (**A**), highlighting the folded (−) and unfolded (+) LRRD (**B**) and MSD (**C**). Different regions and binding sites for VWF-A1 and mAbs are indicated. (**D–G**) Normalized histograms (bar) and their multimodal Gaussian fits (curve) of GPIbα (or GC) unfolding lengths pulled by engaged A1 (**D,E**) or AN51 (**F,G**) in indicated probe–target pairs. Peak values and percentages of unfolding lengths are indicated along with the identified unfolding domains. (**H,I**) Validation of MSD (**H**) and LRRD (**I**) unfolding. The WLC model was fit (curves) to the unfolding force vs. length data (black circles, mean ± s.e.m. of 15–25 measurements per point) from the WM23 vs. platelet experiments where only MSD unfolding was possible (**H**) or A1 vs. GC experiments where only LRRD unfolding was possible (**I**), yielding a contour length of 25.99 ± 0.85 nm or 70.29 ± 3.56 nm, respectively. Overlying on the two panels are corresponding unfolding force vs. length data (red triangles, mean ± s.e.m. of 20–30 measurements per point) from A1 vs. platelet ramped experiments where unfolding of MSD, LRRD or both were all possible, but were segregated into putative MSD (**H**) and LRRD (**I**) unfolding groups based on our decision rules in *Figure 3—source data 1A*. (**J,K**) Mean ± s.e.m. (*n* ≥ 20) of unfolding frequency (**J**) and length (**K**) of LRRD (A1 vs. GC) and MSD (WM23 vs. platelet). Force waveforms indicated as ramped force (1000 pN/s) and clamped force (25 pN) were generated with force-ramp and force-clamp experiment modes respectively. N.D. = Not detected.

The following figure supplement is available for figure 2:

**Figure supplement 1.** Statistical analysis on ramped unfolding length distribution.

Unfolding lengths derived from both signatures were measured from the probe bead position vs. time data (*Figure 1E* insert, 1F and *Figure 1—figure supplement 2*). The lengths of individual ramped unfolding events distributed tri-modally with three subpopulations (*Figure 2D* and *Figure 2—figure supplement 1*; Materials and methods). The first subpopulation coincides with the ramped unfolding length distribution from WM23 vs. platelet experiments (*Figure 2E*, white bars). WM23 binds the MP region below the LRRD (*Figure 2A*), hence could unfold MSD only. The average unfolding force vs. length data from the WM23 experiment was well fitted by the WLC model, yielding a contour length of 25.99 ± 0.85 nm (*Figure 2H*) that matches the previously reported MSD contour length (*Zhang et al., 2015*). The average unfolding force vs. length data from the A1 experiment overlaid well on the same WLC model fit (*Figure 2H*). These results identify the first subpopulation in *Figure 2D* as MSD unfolding.

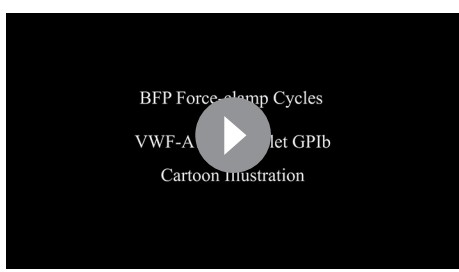

**Video 2.** Force-clamp experiment mode with a bond lifetime event. The video consists of two parts in series. Part I is an animation (produced by Adobe Flash; 12 fps), and part II is a video recording of a representative fluorescence BFP experiment (recorded by a customized LabView program; 25 fps). Both parts show BFP force-clamp measurement cycles. In part I, the synchronized BFP illustration (upper panel), A1–GPIbα interaction (middle panel) and 'Force vs. Time' signal (lower panel) of the same force-clamp cycle with a lifetime event are displayed in parallel. Phases of the BFP cycle are indicated in the lower panel. Part II shows two BFP cycles, which sequentially render a no bond event and a bond lifetime event. The pseudo-color epifluorescence images (acquired at 1 fps) are interpolated and superimposed onto the brightfield images to reflect the real-time intraplatelet $Ca^{2+}$ level (in a progressive sequence: blue, green, yellow, orange and red). Following the long lifetime event, calcium first rapidly elevates and then quickly decays, manifesting an α-type $Ca^{2+}$.

The second subpopulation in *Figure 2D* matches the histogram of ramped unfolding lengths of GC pulled via A1 (*Figure 2E*, blue bars) that ranges from 18–56 nm and peaks at 36 nm (length of leucine-rich repeats 3–6). The average unfolding force vs. length plots derived from the A1 vs. platelet and A1 vs. GC experiments overlaid well on the same WLC model fit (*Figure 2I*). The best-fit contour length (70.29 ± 3.56 nm) matches the length of LRRD, calculated using a 4-Å contour length per residue (*Ju et al., 2015b*). These results identify the second subpopulation in *Figure 2D* as LRRD unfolding.

The third subpopulation in *Figure 2D* can be identified as concurrent unfolding of both MSD and LRRD that occurred within too short a time elapse to be distinguished by our BFP as two separate events, because its maximum unfolding length (85 nm) matches the sum of the observed maximum MSD and LRRD unfolding lengths.

Similar tri-modally distributed ramped unfolding lengths were obtained by using mAb AN51 (epitope mapped to the N-terminal flanking region above LRRD, cf. *Figure 2A*) instead of A1 to pull the platelet GPIbα (*Figure 2F*), and the second subpopulation also matches the ramped unfolding length distribution obtained using AN51 to pull GC (*Figure 2G*, blue bars). These results are expected because the unfolding lengths are determined by the respective primary structures of the LRRD and MSD, and as such should not depend on the 'grabbing handle' used to pull GPIbα. The consistence of the A1 and AN51 results imparts confidence in our identification of the three subpopulations as unfolding of MSD, LRRD, and both, respectively.

Interestingly, the two force waveforms induced unfolding of different GPIbα domains. Clamped forces unfolded only MSD as the lengths of clamped unfolding distribute as a single peak at 20 nm (*Figure 2E,G*, red bars), matching the first subpopulation in *Figure 2D,F*, respectively, regardless of whether platelet GPIbα was engaged by A1 or AN51. Furthermore, unfolding of LRRD in GC was induced only by ramped forces but not clamped forces (*Figure 2J,K*). By comparison, pulling platelet GPIbα via WM23 with both ramped and clamped forces induced MSD unfolding events with similar occurrence frequencies and unfolding lengths (*Figure 2J,K*). These results indicate that MSD can be unfolded by increasing forces as well as constant forces. By comparison, LRRD unfolding requires increasing forces. Some force-clamp cycles (*Figure 1C*; *Video 2*) generated two consecutive unfolding events, one in the ramping and the other in the clamping phase (*Figure 1E*). The respective unfolding lengths of the ramped and clamped unfolding events were 34–55 nm and 13–25 nm that totaled 47–80 nm, agreeing with those of the LRRD, MSD, and MSD+LRRD subpopulations in *Figure 2D,F*. Together, these results provide criteria to determine whether and which GPIbα domain(s) is unfolded (*Figure 3—source data 1A*).

## Force- and ligand-dependent cooperativity between LRRD and MSD unfolding

To characterize the mechanical response of GPIbα, we measured the frequency, force and length of LRRD and MSD unfolding induced by a range of clamped forces exerted on platelet GPIbα or GC by A1WT or a type 2B VWD mutant A1R1450E. The ramped unfolding frequencies of both domains were extremely low at ≤10 pN but increased with the higher levels of clamped forces (*Figure 3A,B*). Interestingly, LRRD, but not MSD, unfolded more frequently when platelet GPIbα (*Figure 3A*) and

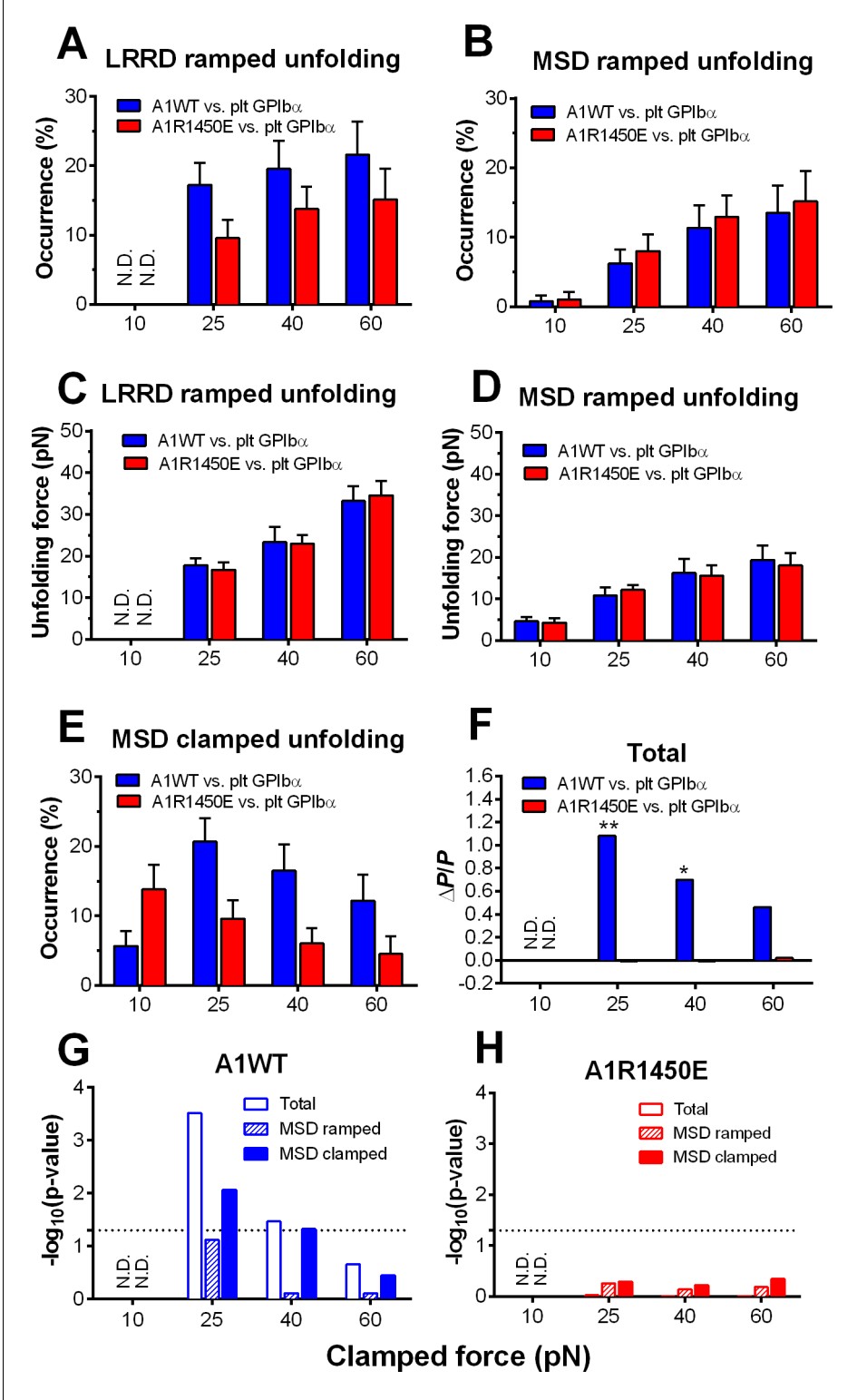

**Figure 3.** Force- and ligand-dependent cooperative unfolding of GPIbα LRRD and MSD. (**A–D**) Frequency (**A,B**) and force (**C,D**) of LRRD (**A,C**) or MSD (**B,D**) unfolding events occurred in the ramping phase induced by pulling via A1WT (blue) or A1R1450E (red) with indicated preset clamped forces. (**E**) Occurrence frequencies of MSD clamped unfolding induced by holding at indicated clamped forces with A1WT or A1R1450E bonds. (**F**) The degree of cooperativity, quantified by $\Delta P/P = P(\text{MSD+LRRD})/[P(\text{MSD}) \times P(\text{LRRD})] - 1$, is plotted vs. clamped force. $P(\text{LRRD})$, $P$ (MSD) and $P(\text{LRRD+MSD})$ are the observed occurrence frequencies of unfolding events of LRRD alone, MSD alone

*Figure 3 continued on next page*

*Figure 3 continued*

and LRRD+MSD, respectively. (**G,H**) Significance of cooperativity assessed by (negative $\log_{10}$ of) p-value of the $\chi^2$ test of the null hypothesis H0: MSD unfolding and LRRD unfolding are independent. The $\chi^2$ test was not performed at 10 pN since under this force LRRD unfolding did not occur and hence no unfolding cooperativity. N. D. = not detected (**A,C**) or not done (**F–H**).

The following source data and figure supplement are available for figure 3:

**Source data 1.** Statistics and cooperativity evaluation of the GPIbα domains unfolding.

**Figure supplement 1.** GC LRRD unfolding occurrence frequencies.

GC (*Figure 3—figure supplement 1*) were pulled by A1WT than A1R1450E. The ramped unfolding forces of both domains increased with the clamped force and were indifferent to whether force was applied via WT or R1450E mutant of A1 (*Figure 3C,D*). In general, a higher force was required to unfold LRRD than MSD. Surprisingly, pulling platelet GPIbα via different ligands generated distinctive MSD clamped unfolding frequency vs. force plots: increasing initially and decreasing after reaching maximal at 25 pN when pulled by A1WT, but decreasing monotonically when pulled by A1R1450E (*Figure 3E*). These data suggest that the mechanoreceptor GPIbα may be able to interpret mechanical cues and discriminate ligands by responding to different force waveforms applied via different ligands with distinct LRRD and MSD unfolding frequencies. In addition, the distinctive force-dependences of two subpopulations of events that we deemed as respective LRRD and MSD unfolding provide further support for our criteria for their identification and classification.

The spatial separation of LRRD and MSD by the >30 nm long MP stalk and the distinctive dependences of their unfolding on the force waveform would seem to favor these two GPIbα domains to unfold independently. This hypothesis predicts that the probability for LRRD and MSD to unfold concurrently should be equal to the product of the respective probabilities for LRRD and MSD to unfold separately. To test this hypothesis, we estimated these probabilities from the observed unfolding occurrence frequencies. At 25 pN, the 34.5% of BFP force-clamp cycles with unfolding events consist of 7.6, 17.2, 6.9, and 2.8% of unfolding of LRRD alone, MSD alone, LRRD and MSD sequentially, and concurrently (*Figure 3—source data 1A*). Significantly, the frequency of observing both LRRD and MSD unfolding in the same binding cycle, calculated by pooling together both cases of two domains unfolding sequentially and concurrently, $P$(MSD+LRRD), is much higher than the product of their respective occurrence frequencies, $P$(MSD)×$P$(LRRD), which is the joint probability for both to unfold assuming that they were independent (*Figure 3—source data 1B*).

These data suggest that the two GPIbα domains may unfold cooperatively, i.e., one domain unfolding may increase the likelihood for the other to unfold. To quantify the degree of such cooperativity, we defined a relative probability difference, $\Delta P/P = [P$(MSD+LRRD) - $P$(MSD)×$P$(LRRD)]/[$P$(MSD)×$P$(LRRD)]. $\Delta P/P > 0$ indicates positive cooperativity between LRRD and MSD unfolding. No cooperativity was observed at 10 pN because this force was insufficient to induce appreciable LRRD unfolding. Pulling with A1WT by a 25 pN clamped force generated high cooperativity, and further increase in force decreased cooperativity (*Figure 3F*). Remarkably, unfolding cooperativity was completely abolished at all forces when applied via the VWD mutant A1R1450E (*Figure 3F*).

We used $\chi^2$ test to determine if the hypothesis that MSD and LRRD unfolded independently should be rejected (Materials and methods). At 25 pN, LRRD unfolding significantly enhanced MSD unfolding (p = $3.09 \times 10^{-4}$). The $\chi^2$ test results are depicted as negative log p-values vs. force plots in *Figure 3G,H* for A1WT and A1R1450E, respectively. Interestingly, significant (p = 0.05, dashed horizontal lines) unfolding cooperativity was observed only for A1WT at 25 and 40 pN. These data show that the cooperativity between LRRD and MSD unfolding is force- and ligand-dependent.

## Model for cooperativity between LRRD and MSD unfolding

To elucidate the mechanism underlying the force- and ligand-dependent unfolding cooperativity, we note that when the MSD unfolding events were separately analyzed according to their occurrence in the ramping or clamping phase, MSD clamped, but not ramped, unfolding was significantly (p= $8.79 \times 10^{-3}$ vs. 0.076 at 25 pN) enhanced by LRRD unfolding (*Figure 3G*), which occurred in the

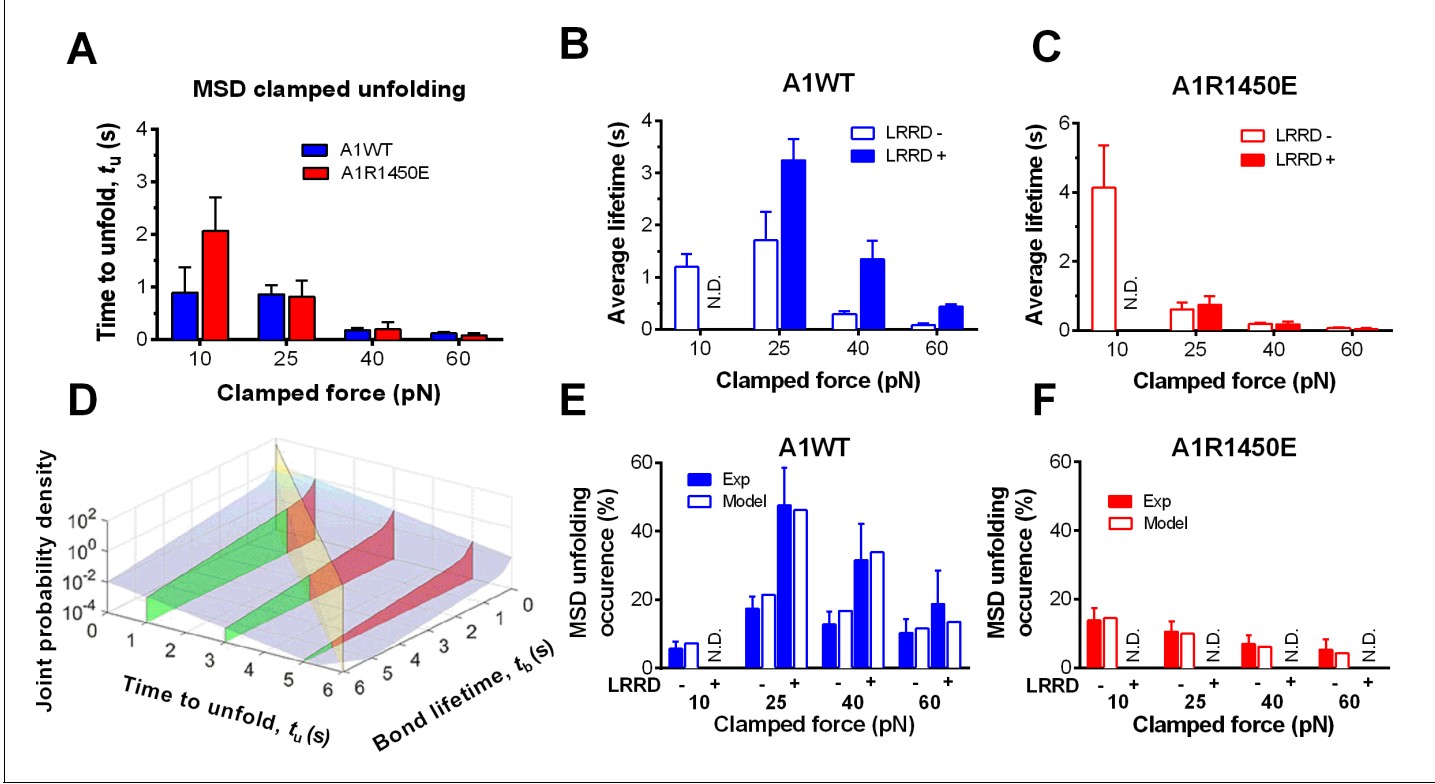

**Figure 4.** LRRD unfolding prolongs A1–GPIbα bond lifetime and facilitates MSD clamped unfolding. (**A–C**) Mean ± s.e.m. of MSD time-to-unfold ($t_u$, **A**) and GPIbα bond lifetimes ($t_b$, **B,C**) with A1WT (blue) or A1R1450E (red) were measured in the clamping phase at different forces in the absence (−) or presence (+) of LRRD unfolding in the same BFP cycle. No LRRD unfolding occurred at 10 pN; hence no bond lifetime was measured under the LRRD+ at this force. (**D**) 3D plot of the surface of joint probability density (z-axis) of GPIbα to dissociate from A1WT at $t_b$ (x-axis) and MSD to unfold at $t_u$ (y-axis) (Materials and methods). Three planes, $t_u$ = 1, 3, and 5 s, under the probability density surface (gray) are shown in green or red, depending on whether they are on the left or right side of the $t_u$ = $t_b$ plane (yellow). (**E,F**) Measured (solid bars) and predicted (open bars) frequency of MSD unfolding events occurred in the clamping phase induced by the indicated force exerted via A1WT (**E**) or A1R1450E (**F**) in the presence (+) or absence (−) of LRRD unfolding in the same BFP cycle. N.D. = not detected. Error bar = s.e.m. estimated by the multinomial distribution of events.

The following source data and figure supplement are available for figure 4:

**Source data 1.** MSD unfolding rates ($k_u$) and the fraction ($w_1$) and off-rates ($k_1$, $k_2$) of GPIbα dissociating from A1WT or A1R1450E under different forces.

**Figure supplement 1.** MSD time-to-unfold distribution for A1WT and 3D probability density surface plot for A1R1450E.

ramping phase only. This dominance of cooperativity by sequential rather than concurrent unfolding suggests a model for LRRD unfolding to impact MSD unfolding, which includes three ideas. The first idea has to do with the MSD time-to-unfold, $t_u$ (cf. *Figure 1F*). Our force-clamp measurements revealed similar $t_u$ values induced by A1WT or A1R1450E pulling (*Figure 4A*). The only exception is at 10 pN where a shorter $t_u$ was induced by A1WT than A1R1450E. This can be explained by their differential bond lifetimes (*Figure 4B,C*). Compared to A1R1450E, the much shorter lifetime of GPIbα bond with A1WT at 10 pN may underestimate $t_u$ because early dissociation of GPIbα would prevent observation of slow MSD unfolding events. This reasoning provides the second idea for our model: MSD clamped unfolding should occur before A1–GPIbα dissociation. The third idea comes from our previous observation (*Ju et al., 2015b*) that LRRD unfolding significantly prolongs GPIbα bond lifetime with A1WT (*Figure 4B*) but not A1R1450E (*Figure 4C*). Combining these three ideas, our model proposes that the A1–GPIbα bond lifetime, regulated by force and prolonged by LRRD unfolding in respective ligand-specific manners, determines the occurrence of MSD clamped unfolding, which, despite its ligand-independent unfolding kinetics, generates a cooperativity pattern that maximizes at the optimal force of 25 pN for A1WT but not for A1R1450E.

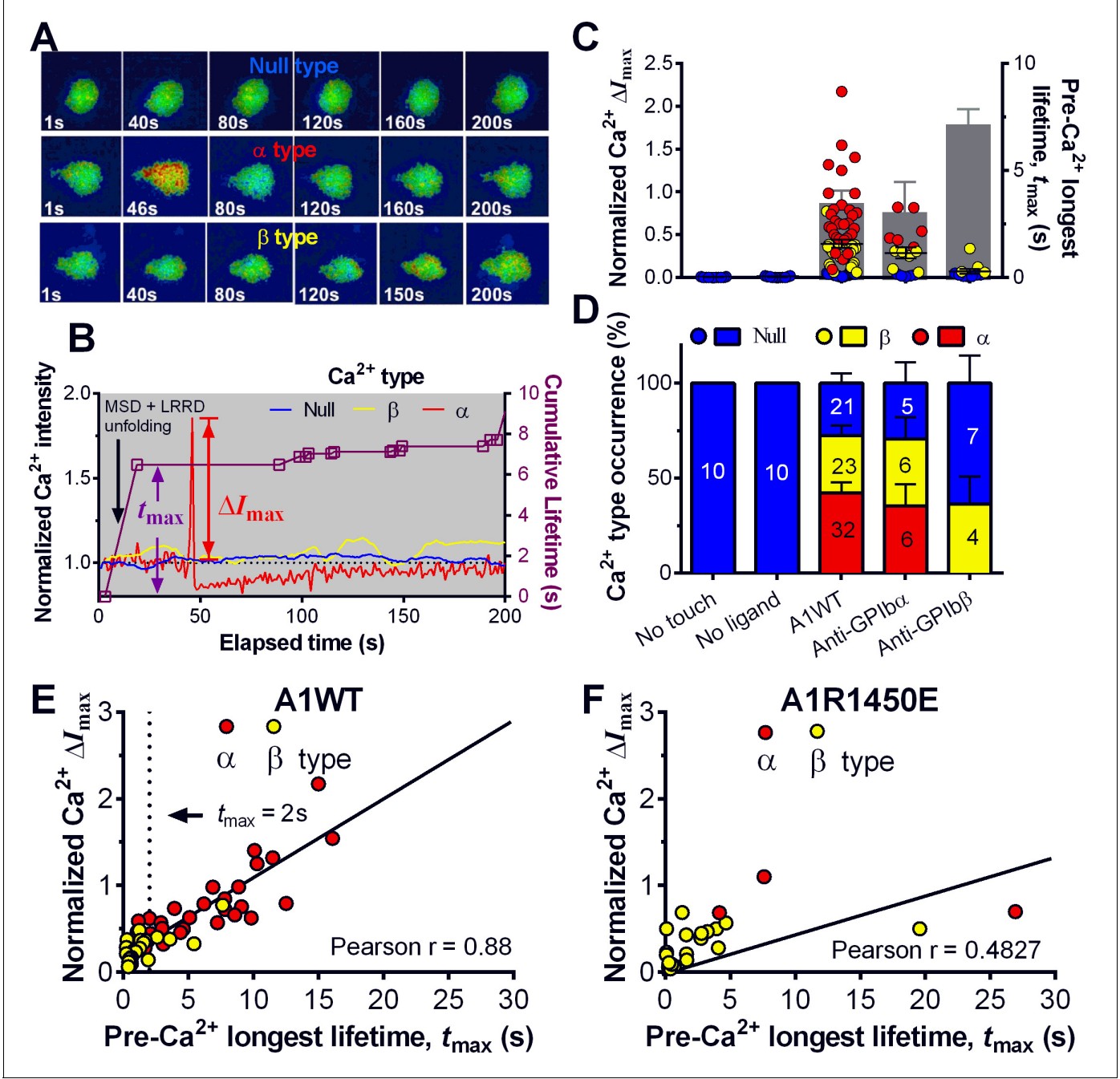

**Figure 5.** Concurrent analysis of single-platelet $Ca^{2+}$ flux and GPIb-mediated single-bond binding at 25 pN clamped force. (**A**) Representative epifluorescence pseudo-colored images of intraplatelet $Ca^{2+}$ of null (top row), $\alpha$- (middle row), and $\beta$- (bottom row) types at indicated times. (**B**) Representative time courses of normalized $Ca^{2+}$ intensity of the null (blue), $\alpha$ (red) and $\beta$ (yellow) types. The concurrent measurement of bond lifetime events (symbol) and the cumulative lifetime (curve) for the platelet exhibiting $\alpha$-type $Ca^{2+}$ is overlaid. The pre-$Ca^{2+}$ longest lifetime ($t_{max}$) and the maximum intensity increase of the $\alpha$-type $Ca^{2+}$ ($\Delta I_{max}$) are indicated. The time when a concurrent LRRD and MSD unfolding event occurred is indicated by the arrow. (**C,D**) Individual $\Delta I_{max}$ values and their mean ± s.e.m. (points, left ordinate) and mean ± s.e.m. of $t_{max}$ (gray bars, right ordinate) (**C**) and fractions (**D**) of $Ca^{2+}$ types triggered by different stimulations. Each point in (**C**) represents results from one platelet and the numbers of platelets in each column are indicated in the corresponding bar in (**D**), with matched colors to indicate $Ca^{2+}$ types. (**E,F**) Scattergraphs of $\Delta I_{max}$ vs. $t_{max}$ for A1WT (**E**) and A1R1450E (**F**). The solid lines are linear fits to respective data with corresponding Pearson coefficients indicated. The null-type $Ca^{2+}$ data was excluded in the analysis.

The following figure supplements are available for figure 5:

**Figure supplement 1.** Concurrent analysis of single-platelet $Ca^{2+}$ flux and single-bond dissociation from GPIb at 25 pN clamped force.

*Figure 5 continued on next page*

*Figure 5 continued*

**Figure supplement 2.** Specificity-sensitivity analysis of optimal threshold.

To formulate the model mathematically, we multiplied the respective probability densities of the exponentially distributed MSD time-to-unfold ($t_u$) (*Figure 4—figure supplement 1A*) and the dual-exponentially distributed lifetime ($t_b$) of GPIbα bonds with A1WT or A1R1450E (*Ju et al., 2013*) to construct a joint probability density surface over the $t_u$-$t_b$ plane (*Figure 4D* and *Figure 4—figure supplement 1B*). The predicted MSD clamped unfolding probability is the volume under this surface over the region $0<t_u<t_b<\infty$ (Materials and methods). When the model was tested against experiment, not only did the calculated force-dependent MSD unfolding frequency match the biphasic pattern for A1WT (*Figure 4E*) and the monophasic pattern for A1R1450E (*Figure 4F*), but it also compared well with the observed occurrence frequencies numerically at all forces. Remarkably, the model predicts both the quantitative enhancement of MSD unfolding by LRRD unfolding for A1WT and the lack of enhancement for A1R1450E without a single freely adjustable fitting parameter. The excellent agreement between theory and experiment has provided strong support for our model and explained the data in *Figure 4E,F*.

## Platelet signaling induced by mechanoreception via a single GPIbα

Platelet translocation on VWF signals through GPIbα to induce $Ca^{2+}$ fluxes (*Mazzucato et al., 2002*; *Nesbitt et al., 2002*). We optimized the fluorescence BFP (fBFP) method (*Chen et al., 2015*; *Liu et al., 2014*) for single-platelet calcium imaging and studied how platelet signaling was triggered by GPIbα mechanoreception via a sequence of intermittent single bonds under a range of clamped forces. The $Ca^{2+}$ signals over the 200-s observation window of repeated platelet contact cycles were classified into three types (*Figure 5A,B*): i) null-type, featured by a basal trace with a maximum $Ca^{2+}$ intensity increase (normalized by its initial value) $\Delta I_{max}<0.05$; ii) α-type, featured by an initial latent phase followed by a spike (mostly $\Delta I_{max}>0.5$) with a quick decay (*Video 2*); iii) β-type, featured by fluctuating signals around the baseline or gradually increasing signals to an intermediate level (mostly $\Delta I_{max}<0.5$) followed by a gradual decay to baseline (*Video 3*). The null type reflects the baseline with background noise, while the α- and β-types match the previous characterization of platelet internal $Ca^{2+}$ release triggered by VWF–GPIbα bonds measured in flow chamber experiments (*Mazzucato et al., 2002*). For each platelet, the calcium trace was overlaid with the sequential binding events, bond lifetimes, and their accumulation over the repeated platelet binding cycles (*Figure 5B* and *Figure 5—figure supplement 1A,B*).

Pulled by a 25 pN clamp force, A1WT–GPIbα bonds triggered much higher $\Delta I_{max}$ than controls (*Figure 5C*), showing 28, 42, and 30% of null-, α- and β-types, respectively (*Figure 5D*). Control experiments that merely held aspirated platelets or contacted them by beads without coating any ligand showed null-type $Ca^{2+}$ only (*Figure 5C,D*). The α-type $Ca^{2+}$ could also be triggered by pulling GPIbα with AN51 but not with an anti-GPIbβ mAb (*Figure 5C,D* and *Figure 5—figure supplement 1A,B*), despite that GPIbβ is tightly connected to GPIbα within one GPIb complex and has been postulated to play a role in signaling through GPIb (*Strassel et al., 2006*). These data demonstrated the necessity of GPIbα engagement to trigger intraplatelet $Ca^{2+}$

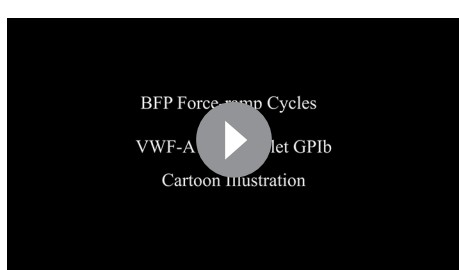

**Video 3.** Force-ramp experiment mode with a bond rupture event. Similar to *Video 2*, this video consists of two parts in series. In part I, the synchronized BFP illustration (upper panel), A1–GPIbα molecular interaction (middle panel) and 'Force vs. Time' signal (lower panel) of the same force-ramp cycle with a ~65 pN rupture force event are displayed in parallel. Part II shows two BFP cycles, which sequentially render a no bond event and a bond rupture event. After the bond rupture event, low level calcium mobilization occurs right away, namely the β-type $Ca^{2+}$.

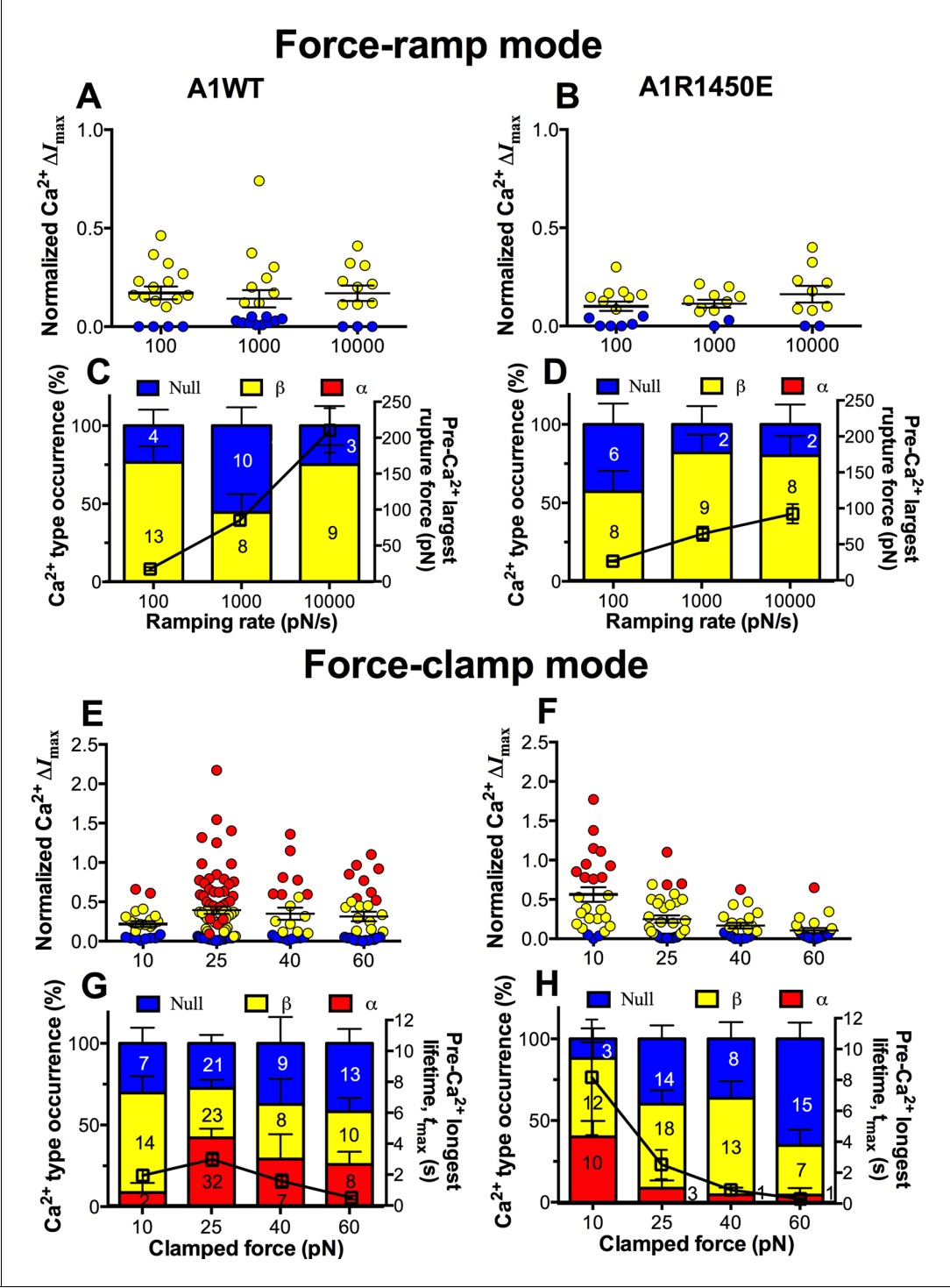

**Figure 6.** GPIbα can sense different force waveforms and discriminate different ligands. (A–D) Force-ramp fBFP experiment mode. Individual $\Delta I_{max}$ values and their mean ± s.e.m. (A,B, points), $Ca^{2+}$ types (C,D, stacked bars, left ordinate), and mean ± s.e.m. of pre-$Ca^{2+}$largest rupture force (C,D, black square, right ordinate) are plotted vs. force ramping rate for A1WT (A,C) or A1R1450E (B,D). (E–H) Force-clamp fBFP experiment mode. Individual $\Delta I_{max}$ values and their mean ± s.e.m. (E,F, points), $Ca^{2+}$ types (G,H, stacked bars, left ordinate), and mean ± s.e.m. of pre-$Ca^{2+}$ longest lifetime (G,H, black square, right ordinate) are plotted vs. clamped force for A1WT (E,G) or A1R1450E (F,H). Each point in (A,B,E,F) represents results from one platelet and the numbers of platelets in each column are indicated in the corresponding bar in (C,D,G,H), with matched colors to indicate $Ca^{2+}$ types. Error bar in (C,D,G,H) represents s.e.m. estimated by the multinomial distribution of events.

and agree with the previous report that α-type $Ca^{2+}$ peaks occur when platelets are transiently arrested in the whole blood flow (*Mazzucato et al., 2002*; *Nesbitt et al., 2002*)

The concurrent measurements of A1–GPIbα binding kinetics and intraplatelet $Ca^{2+}$ allowed us to determine the pre-$Ca^{2+}$ bond lifetimes (*Figure 5B*), enabling single platelet correlative analysis of binding and signaling. Using the normalized maximum calcium intensity $\Delta I_{max}$ to represent the $Ca^{2+}$ level, we compared its correlations with three statistics of A1WT–GPIbα bond lifetimes occurred prior to calcium onset. We found that $\Delta I_{max}$ correlates best with the pre-$Ca^{2+}$ longest bond lifetime $t_{max}$ (*Figure 5E*), similarly well with the pre-$Ca^{2+}$ cumulative lifetime $\sum t_i$ (*Figure 5—figure supplement 1C*), but poorly with the pre-$Ca^{2+}$ average lifetime $<t>$ (*Figure 5—figure supplement 1D*). Careful examination of many overlaid calcium and bond lifetime traces, as exemplified in *Figure 5B*, revealed that the $\sum t_i$ values are generally dominated by the $t_{max}$ values, which are usually much longer than the rest of the pre-$Ca^{2+}$ bond lifetimes and are immediately followed by the calcium onset before observing additional shorter bond lifetimes. In other words, for each platelet usually $\sum t_i$ could be approximated by $t_{max}$ but $<t>$ is of a smaller and variable value. This observation explains why calcium correlates equally well with $t_{max}$ and $\sum t_i$ but not with $<t>$. Importantly, these results also suggest that a single long-lived GPIbα bond is sufficient to trigger $Ca^{2+}$ in a platelet. This assertion has been further supported by the parallel analysis of the data for A1R1450E. Although for R1450E the $t_{max}$ value was significantly shorter and the α-type $Ca^{2+}$ population was greatly reduced, the $\Delta I_{max}$ still showed similar correlation with $t_{max}$ (*Figure 5F*). Thus the pre-$Ca^{2+}$ longest bond lifetime correlates the $Ca^{2+}$ strength and type.

## GPIbα discriminates ligands and shows different force-dependent calcium responses

We next asked whether the mechanoreceptor GPIbα is capable of sensing differences in the force waveform and discriminating the ligand through which force is applied. We first performed force-ramp experiment to generate a wide range of rupture forces using three ramping rates: 100, 1000 and 10,000 pN/s. However, only low levels of β-type $Ca^{2+}$ were resulted (*Figure 6A,B*), showing no correlation with the largest rupture force prior to calcium onset (*Figure 6C,D*, right ordinate), regardless of whether platelets were tested by A1WT (*Figure 6A,C*) or A1R1450E (*Figure 6B,D*). In sharp contrast, much higher levels of $Ca^{2+}$ of α- and β-types were induced by clamped forces applied to GPIbα via either A1WT (*Figure 6E,G*) or A1R1450E (*Figure 6F,H*) despite their much lower levels than the rupture forces seen in the force-ramp experiments (*Figure 6C,D*). Concurrently, the longest of A1–GPIbα bond lifetimes that occurred prior to $Ca^{2+}$ onset was measured on each platelet and averaged over all platelets in each group. This pre-$Ca^{2+}$ longest bond lifetime, $t_{max}$, exhibited catch-slip bond behavior for A1WT and slip-only bond behavior for A1R1450E (*Figure 6G, H*, right ordinate), just as the corresponding average bond lifetimes previously measured regardless of the intraplatelet calcium (*Ju et al., 2013*; *Yago et al., 2008*). Remarkably, the force-dependent pattern of calcium signals matched that of the pre-$Ca^{2+}$ longest bond lifetimes for both A1WT and A1R1450E. The ligand-independent positive correlation of $Ca^{2+}$ signal with $t_{max}$ is consistent with a previously observed inverse correlation between the cytosolic $Ca^{2+}$ level and the translocation velocity of platelets on immobilized VWF (*Nesbitt et al., 2002*). This is expected because the platelet translocation velocity is an inverse metric of VWF–GPIbα bond lifetime (*Ju et al., 2013*; *Yago et al., 2008*).

## Interplay among GPIbα engagement duration, domain unfolding and signal initiation

The findings that durable force is important to both MSD unfolding and $Ca^{2+}$ triggering prompted us to investigate the relationship between GPIbα domain unfolding and platelet signal initiation. We segregated the $Ca^{2+}$ data generated by a 25 pN clamped force on A1WT–GPIbα bonds according to whether or not and, if so, which domain(s) was (were) unfolded prior to calcium onset. Platelets whose tests contained no unfolding event showed short $t_{max}$ and low calcium of β- and null-types (*Figure 7A*). Platelets whose tests contained at least one pre-$Ca^{2+}$ MSD unfolding event but no LRRD unfolding showed slightly longer $t_{max}$ and higher $Ca^{2+}$ of mostly α-type. By comparison, only β-type $Ca^{2+}$ was observed in the rare (2.6%) cases where LRRD unfolded but MSD did not. Since in these cases the $t_{max}$ values were much longer, this data excludes $t_{max}$ to be the direct determining

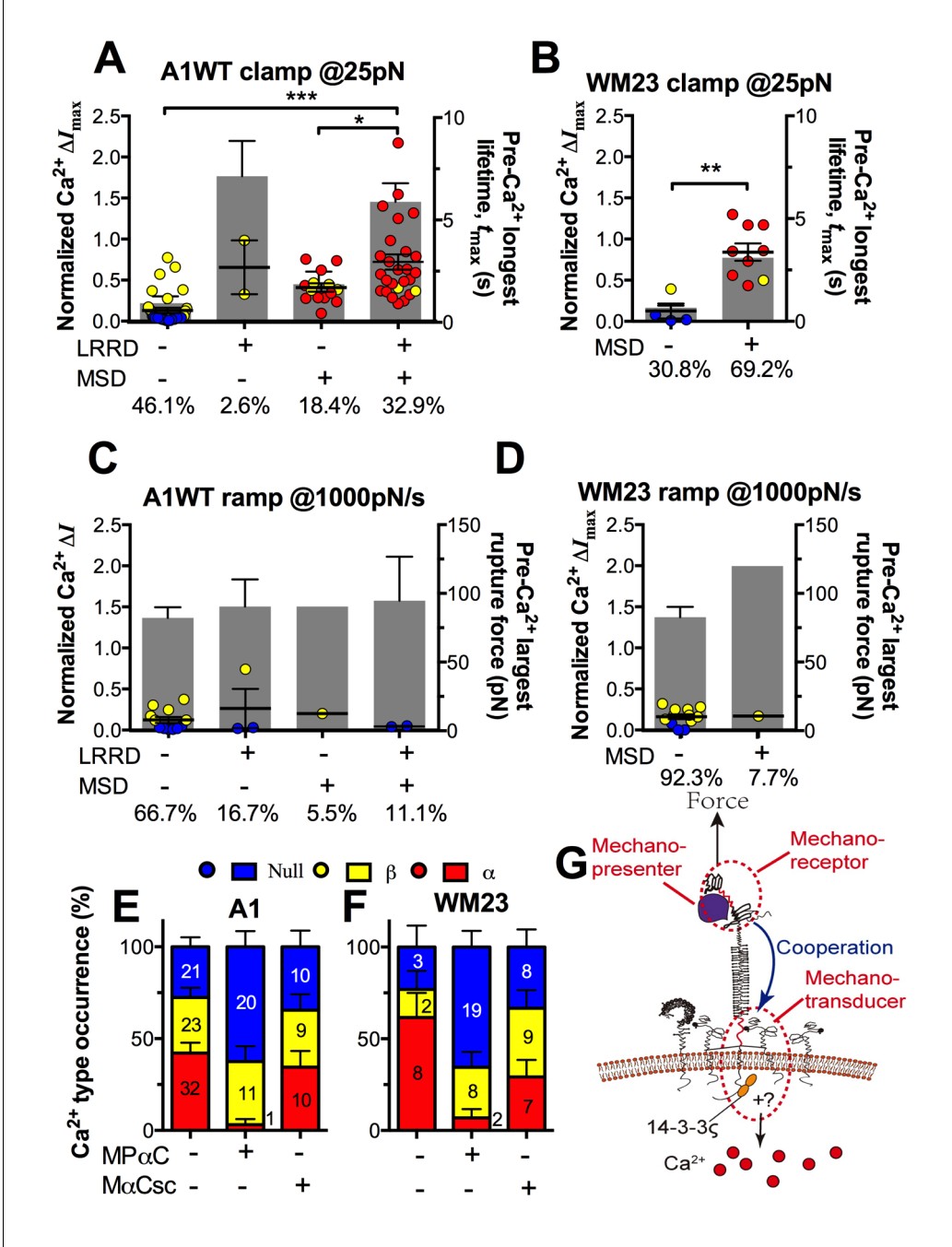

**Figure 7.** Correlation between GPIbα domain unfolding and Ca²⁺ triggering at 25 pN clamped force. (**A–D**) Individual $\Delta I_{max}$ values and their mean ± s.e.m. (points, left ordinate) in platelets triggered by A1 (**A,C**) or WM23 (**B, D**) binding, which were segregated into groups with (+) or without (−) unfolding of LRRD and/or MSD. Each point represents measurement from a platelet. The frequency of each unfolding combination to occur was indicated. (**A, B**) Data obtained from 25 pN force-clamp experiments. Corresponding $t_{max}$ (gray bars, right ordinate) were overlaid with $\Delta I_{max}$. (**C,D**) Data obtained from 1000 pN/s force-ramp experiments. Corresponding pre-Ca²⁺ largest rupture force (gray bars, right ordinate) were overlaid with $\Delta I_{max}$. (**E,F**) Percentage of total events of three Ca²⁺ types in platelets in the same experiments as in (**A**) and (**B**) (left bars) as well as additional experiments performed in the presence of MPαC (middle bars) or MαCsc (right bars). Error bar = s.e.m. estimated from the multinomial distribution of events. (**G**) A postulated model of GPIbα-mediated mechanosensing. Force applied via VWF-A1 induces GPIbα LRRD and MSD unfolding. GPIbβ head domain binds to the unfolded MSD and causes the dissociation of its cytoplasmic tail from GPIbα-associated 14-3-3ζ, which transduces signals across the platelet

*Figure 7 continued on next page*

*Figure 7 continued*

membrane and further downstream, finally leading to α-type $Ca^{2+}$. Each step of the mechanosensing process is indicated.

The following figure supplements are available for figure 7:

**Figure supplement 1.** Comparison of GPIbα bond lifetime and MSD clamped unfolding by A1WT and WM23.

**Figure supplement 2.** Model of GPIb-mediated platelet mechanosensing.

parameter for the $Ca^{2+}$ type. Remarkably, the group with pre-$Ca^{2+}$ unfolding of both LRRD and MSD exhibited long $t_{max}$ and high $Ca^{2+}$ of mostly α-type.

Consistent results were obtained using WM23 to pull GPIbα to bypass LRRD unfolding (cf. *Figure 2A*), showing significantly longer $t_{max}$ and higher calcium for platelets with than without a MSD unfolding event and a clear α- vs. β-type signal distinction between them (*Figure 7B*). The higher α-type $Ca^{2+}$ triggering efficiency of WM23 than A1 (compare MSD+ columns in *Figure 7A and B*) may be explained, at least in part, by the slower kinetics of GPIbα dissociation from WM23 than A1, which generated 70% more bond lifetime events by contacting a platelet for 200 s with WM23 than A1 (*Figure 7—figure supplement 1A*). The bonds were also 74% more durable (*Figure 7—figure supplement 1B*), resulting in a slightly higher (although not significant) MSD clamped unfolding probability per lifetime event for WM23 (*Figure 7—figure supplement 1C*). The expected number of MSD clamped unfolding over a 200-s experimental period, calculated as the product of the number of lifetime events and the unfolding probability per lifetime event, was significantly higher for WM23 than A1 (*Figure 7—figure supplement 1D*). The probability of platelet to have at least one MSD unfolding, calculated as 1– (1– unfolding probability per lifetime event)ˆ(# lifetime events), is 40% and 60% for A1 and WM23, respectively (*Figure 7—figure supplement 1E*), close to the experimental results (51% vs. 69%, *Figure 7A,B*).

Interestingly, despite their high levels, ramped forces generated very few MSD unfolding events and triggered only null/β- but not α-type $Ca^{2+}$ regardless of whether A1WT or WM23 was used to pull (*Figure 7C,D*). Together, these data indicate that both force-induced MSD unfolding and bond lifetime are necessary for inducing α-type $Ca^{2+}$ signal.

Using a sensitivity-specificity analysis (*Figure 5—figure supplement 2A,B*) that slides a putative threshold through the $t_{max}$ vs. $Ca^{2+}$ type data, we found $t_{max}>2$ s to be the best predictor for A1WT to trigger α- rather than β-type $Ca^{2+}$ (*Figure 5E*, dashed lines), which agrees with the fact that the 2 s threshold is much shorter than the average $t_{max}$ of α-type $Ca^{2+}$, but exceeds that of β-type $Ca^{2+}$ and MSD time-to-unfold (*Figure 5—figure supplement 2C*). Thus, a longer-lived pre-$Ca^{2+}$ bond favors MSD unfolding, thereby triggering α-type $Ca^{2+}$; otherwise, it only triggers β-type $Ca^{2+}$. Together, our data suggests separate roles of LRRD and MSD unfolding in GPIbα signaling, with the former intensifying the $Ca^{2+}$ level and the latter determining the $Ca^{2+}$ signal type .

## Perturbing cytoplasmic association of GPIbα with 14-3-3ζ inhibits mechanotransduction

To understand GPIbα-mediated mechanosensing requires analysis of not only ligand binding and domain unfolding in the extracellular segment of GPIbα, but also events in its cytoplasmic region. 14-3-3ζ is a cytoplasmic protein that has direct association with both GPIbα and GPIbβ C-termini (*Calverley et al., 1998*) and regulates GPIb signal transduction(*Dai et al., 2005*). To investigate the role of 14-3-3ζ in GPIbα-mediated $Ca^{2+}$ signaling, we perturbed the system with a myristoylated peptide (MPαC) that mimics the 14-3-3ζ binding sequence of GPIbα, thereby blocking the association of 14-3-3ζ with GPIbα cytoplasmic tail. Consistent with the previously reported signaling inhibition effect (*Dai et al., 2005*; *Yin et al., 2013*), MPαC reduced the fraction of α-type $Ca^{2+}$ from 34 to 3% without affecting β-type $Ca^{2+}$, whereas a scramble peptide MαCsc had no effect (*Figure 7E*). Similar results were obtained by pulling GPIbα via WM23 on platelets (*Figure 7F*). Thus, GPIb–14-3-3ζ association, a biochemical event, is crucial for the transduction of MSD unfolding, a mechanical

event, into intracellular signals. These observations indicate that GPIb–14-3-3ζ serves, at least in part, as a mechanotransducer (*Figure 7G*).

## Discussion

The mechanoreception of GPIbα has been supported by direct observations of transient intracellular $Ca^{2+}$ spike (termed type α/β peak) upon platelet translocation on VWF under flow (*Mazzucato et al., 2002*; *Nesbitt et al., 2002*). However, many questions remain. Using fBFP real-time single-bond, single-platelet analysis of force-regulated ligand binding kinetics, receptor unfolding dynamics and intraplatelet calcium mobilization, we have: 1) identified, characterized and mathematically modeled the force- and ligand-dependent cooperativity between LRRD and MSD unfolding (*Figure 7G*); 2) defined an optimal magnitude and threshold duration of clamped force for platelet signal initiation via a single GPIbα bond (*Figure 7—figure supplement 2*); 3) uncovered a mechanopresentation defect in a type 2B VWD mutant A1R1450E; 4) delineated the interplay among ligand engagement, GPIbα domain unfolding and signal triggering; and 5) revealed inhibition of GPIbα mechanotransduction by perturbing its cytoplasmic association with 14-3-3ζ.

It is an interesting yet challenging problem to define the minimum mechanical stimulation for inducing signal transduction. We demonstrated that a single A1–GPIbα bond can induce calcium in a platelet without clustering GPIb by multimeric ligands. Specifically, in 83% of the cases where α-type $Ca^{2+}$ was triggered, only a single MSD unfolding event was observed before the $Ca^{2+}$ onset. Of these, 43% had only one pre-$Ca^{2+}$ bond lifetime event. Thus, pulling a single GPIbα by a 25 pN force for >2 s to unfold MSD once is necessary and sufficient to induce α-type $Ca^{2+}$ signals. By comparison, to trigger calcium in a naïve $CD8^+$ T lymphocyte requires a sequence of intermittent bonds with a total of >10 s lifetimes under 10 pN accumulated in the first 60 s of contacts between T cell receptors and agonist peptide-major histocompatibility complex molecules (*Liu et al., 2014*). In both cases a threshold of force duration is required, as ramped forces without a clamp phase are unable to trigger appreciable levels of α-type $Ca^{2+}$ regardless of its magnitude.

The binding defects of VWF-A1 with type 2B VWD mutations have long been recognized (*Ruggeri, 2004*). A recent study has shown a type 2B mutation, A1V1316M, causes additional signaling defects (*Casari et al., 2013*). We showed that another type 2B mutation, A1R1450E, also has signaling defect. Interestingly, the defect to induce calcium has the same root as the binding defect, namely, the conversion of the wild-type catch-slip bond to the mutant slip-only bond (*Ju et al., 2013*; *Yago et al., 2008*). Consequently, force exerted on GPIbα by A1R1450E is less able to unfold LRRD, lasts shorter at 25 pN to unfold MSD less frequently, does not generate unbinding cooperativity between the LRRD and MSD, and induces lower level and frequency of α-type $Ca^{2+}$ at >10 pN. Thus, the mechanical requirements for signal induction manifest as force-dependencies of VWF–GPIbα bond lifetime, MSD unfolding frequency, unfolding cooperativity, and $Ca^{2+}$ level/type that display similar patterns for the same A1 construct (WT or R1450E) but different patterns between A1WT and A1R1450E. These findings show that the GPIbα mechanoreceptor can discriminate ligands and shed light to the biophysical mechanisms of type 2B VWD.

Our new data on the interplay among VWF binding, GPIbα unfolding, and $Ca^{2+}$ signaling have provided new insights into the inner workings of the A1–GPIb–14-3-3ζ molecular assembly (*Figure 7*; *Video 1*). By residing in the juxtamembrane stalk region, the MSD has been shown to be mechanosensitive, and hypothesized to play a role in activating platelets (*Zhang et al., 2015*). In the present work, we found that MSD unfolding is required to trigger α-type $Ca^{2+}$, showing that this extracellular mechanical event is necessary for transduction of the information embedded in the force waveform into intracellular biochemical signals via the 14-3-3ζ connection (*Figure 7G*). By overlapping with the ligand-binding site, the LRRD can feel the structural variation in the A1 and respond with an altered unfolding frequency and changed bond lifetime. Importantly, LRRD unfolding prolongs A1–GPIbα bond lifetime to facilitate MSD unfolding, thereby increasing the frequency of α-type $Ca^{2+}$ and its level (*Figure 7—figure supplement 2*). Thus, our study has elucidated part of a mechanosensor that includes three components: 1) a MSD in the juxtamembrane region whose conformational change results in a binary decision of $Ca^{2+}$ type, 2) a LRRD in the ligand-binding region whose conformational change leads to continuous alterations in ligand-binding duration, signal level and fractions of different signal types, and 3) a MP stalk that transmits force over a distance and provides coupling between the two unfoldable domains. The differential unfolding behaviors of the LRRD and MSD in

response to distinct force waveforms provide a simple mechanical mechanism for unfolding cooperativity, by setting the response order such that LRRD unfolds first during force ramp to give more time for MSD to unfold during force clamp. These principles may be helpful for design of a generic mechanosensor, e.g., using synthetic biology approaches.

In addition to an increased MSD unfolding frequency, the cooperativity between LRRD and MSD unfolding may manifest as an increased LRRD unfolding extent. This has been suggested by the observation that the values of the first two peaks (20 and 36 nm) do not add up to that of the third (65-70 nm) in the unfolding length histograms. We note that the observed maximum unfolding length from the GC experiments (56 nm) is smaller than the calculated contour length of LRRD (70 nm). Since LRRD consists of 8 leucine-rich repeats, this suggests that only some but not all of the repeats were unfolded in any given observation. Thus, the LRRD dataset contains mixed populations of partial unfolding events. By comparison, the MSD dataset may be a more uniform population as the observed maximum unfolding length from the WM23 experiments (27 nm) matches the calculated contour length of MSD (26 nm). These considerations suggest possible explanations for the observation that events in which both LRRD and MSD unfold generate more length than the sum of lengths generated from events in which either LRRD or MSD unfolds: LRRD unfolding events with a higher number of unfolded leucine-rich repeats may facilitate MSD unfolding more effectively than those with a lower number of unfolded leucine-rich repeats. Alternatively, MSD unfolding, once happens, may induce more leucine-rich repeats in LRRD to unfold. Note that these two mechanisms are not mutually exclusive.

Studies in mechanosensitive ion channels and enzymes have provided knowledge on how biomolecules respond to force and transduce mechanical stimulations into biochemical signals. For example, ion channels open and close in response to stress within the lipid bilayer or force within a protein link that can do work on the channel and stabilize its state (*Sukharev and Sachs, 2012*). Mechanosensitive enzymes or substrates, such as vinculin (*del Rio et al., 2009*; *Grashoff et al., 2010*) or A2 domain of VWF (*Wu et al., 2010*; *Zhang et al., 2009b*), change conformations in response to forces to expose a cryptic site to enable enzymatic reaction. By comparison, for the system studied herein, force signals are received by the receptor via ligand interaction, hence mediated by their binding kinetics. The process of transducing the extracellular mechanical events (i.e., LRRD and/or MSD unfolding) across the cell membrane is likely mechanical rather than chemical (i.e., ion influx).

The study of GPIb mechanosensing may help understand how mechanical force regulates platelet thrombotic functions. For example, in response to shear gradients resulting from flow perturbations, discoid platelets aggregate rapidly in a manner independent of agonist activation pathways (*Jain et al., 2016*; *Nesbitt et al., 2009*; *Yong et al., 2011*). This intriguing phenomenon has significant implication in atherothrombosis and medical device thrombotic fouling. Other than the requirement for VWF–GPIbα binding, the underlying mechanism of such purely force-induced platelet thrombosis remains elusive. Here LRRD unfolding may play a role because it requires an increasing force (resembles shear gradient) and strengthens VWF–GPIbα bonds (*Ju et al., 2015b*) (*Figure 7—figure supplement 2C*). In addition, our findings may have broad implications since LRRD is a common structure shared by many adhesion and signaling receptors, e.g., toll-like receptors (*Bella et al., 2008*).

## Materials and methods

### Proteins and peptides

Recombinant monomeric VWF-A1 (residues 1238–1471) WT and type 2B mutant R1450E (*Cruz et al., 2000*; *Morales et al., 2006*) generated by E.coli were gifts of Miguel A. Cruz (Baylor College of Medicine, Houston, TX). Glycocalicin was purified from outdated platelets (*Fox et al., 1988*). Three anti-GPIbα mAbs were used: AK2 (Abcam, Cambridge, MA), AN51 (Millipore, Billerica, MA) and WM23 (a gift from Michael Berndt, Curtin University, WA, Australia and Renhao Li, Emory University, Atlanta, GA). Anti-GPIbβ mAb LS-B3174 was purchased (LifeSpan BioSciences, Seattle, WA). Anti-VWF-A1 mAb 6G1 was a gift from Michael Berndt. Myristoylated peptides (MPαC, C13H27CONH-SIRYSGHpSL) and myristoylated scrambled control peptide (MαCsc, C13H27CONH-LSISYGSHR) were produced as previously described (*Dai et al., 2005*; *Yin et al., 2013*).

## Red blood cells (RBCs) and platelets

Human RBCs and platelets were collected abiding a protocol approved by the Institute Review Broad of Georgia Institute of Technology. RBCs were prepared as previously described (*Ju et al., 2013a*). To obtain fresh discoid human platelets, whole blood was drawn slowly from a vein of a healthy volunteer to fill in a 3 ml syringe preloaded with 0.5 ml ACD buffer (6.25 g sodium citrate, 3.1 g citric acid anhidrous, 3.4 g D-glucose in 250 ml deionized $H_2O$, pH 6.7). The whole blood was centrifuged at 150 g for 15 min without brake. Platelet-rich plasma was extracted and centrifuged at 900 g for another 10 min. The platelet pellet was resuspended into Hepes-Tyrode buffer (134 mM NaCl, 12 mM $NaHCO_3$, 2.9 mM KCl, 0.34 mM sodium phosphate monobasic, 5 mM HEPES, and 5 mM glucose, 1% bovine serum albumin (BSA), pH 7.4). For $Ca^{2+}$ imaging experiments, isolated platelets were incubated with Fura-2-AM (Life Technologies, Grand Island, NY) at 30 µM for 30 min. For treatment with MPαC or MαCsc, the peptide pre-dissolved in DMSO was resuspended into the platelet suspension to reach a final concentration of 25 µM and incubated at 37°C for 30 min.

## Functionalization of glass beads

A1WT, A1R1450E and antibodies were pre-coupled covalently with maleimide-PEG3500-NHS (MW ~3500; JenKem, TX). As previously described (*Ju et al., 2013a, 2015a*), the modified proteins were then mixed with streptavidin (SA)-maleimide (Sigma-Aldrich, St. Louis, MO) in carbonate/bicarbonate buffer (pH 8.5) and together linked to silanized borosilicate beads (Thermo Fisher Scientific, Waltham, MA) in phosphate buffer (pH 6.8). Site densities of ligands on beads were measured using the previously described flow cytometry method (*Ju et al., 2015a*).

## Fluorescence biomembrane force probe (fBFP)

Our fBFP was developed to simultaneously measure the binding kinetics of single receptor–ligand bonds (*Ju et al., 2015a, 2013, 2015c*) and the mechanics of single protein conformational changes (*Chen et al., 2012*; *Ju et al., 2015b*), as did our original BFP, and receptor-initiated intracellular signaling with a concurrent fluorescent imaging module (*Chen et al., 2015*; *Liu et al., 2014*). Bond formation, force application, receptor conformational change, and bond dissociation were enabled and monitored in controlled BFP cycles of a few seconds each. Intraplatelet calcium fluxes were ratiometracally imaged as a signaling readout.

In a BFP cycle, the platelet was driven to approach, impinge and hold the probe with a 20–30 pN compressive force for a contact time of 2 s to allow for bond formation, and then retract (ramp) for bond detection (*Figure 1C* and *Figure 1—figure supplement 1A i–iv*). Displacement of the probe bead was tracked, which reflected the force exerted on it. During the ramping phase, a bond event was signified by a tensile force signal (*Figure 1—figure supplement 1C,D*), while no tensile force was detected in a no-bond event (*Figure 1—figure supplement 1B*). Bond and no-bond events were enumerated to calculate an adhesion frequency in 50 repeated cycles for each bead and platelet pair. At least 3 bead–platelet pairs were measured and their adhesion frequencies were used to calculate mean ± s.e.m. values. To define the minimum requirement for GPIb mechanoreception, adhesions were adjusted to be infrequent (<20%) by titrating the densities of randomly distributed A1 and mAb on the probe beads (*Figure 1D*). This ensured that most (>89%) platelet–bead binding events were mediated by non-clustered single-bonds (*Chesla et al., 1998*).

To quantify intraplatelet $Ca^{2+}$ mobilization, we used ratiometric imaging with a light source that alternates two excitation wavelengths (340 ± 10 nm to excite $Ca^{2+}$-bound Fura-2, and 380 ± 10 nm to excite $Ca^{2+}$-free Fura-2). The emission light from the excited Fura-2 (with or without $Ca^{2+}$ binding) was captured by a fluorescence camera. To maintain the physiological temperature (37°C) inside the cell chamber, a custom-designed temperature control system made in house was integrated into the fBFP. Details about the $Ca^{2+}$ imaging analysis and temperature control have been previously described (*Chen et al., 2015*; *Liu et al., 2014*).

## Force-clamp and force-ramp experiment modes

In the force-clamp mode, the target pipette was driven to repeatedly contact the probe bead for 2 s and retract at a constant speed (3.3 µm/s). Multiplying the BFP spring constant (0.3 pN/nm), this would translate to a linearly increasing force at a constant ramping rate (1000 pN/s). Upon detection of bond event, a feedback loop controls the retraction so that it would be paused at a desired

clamped force (10, 25, 40 and 60 pN) to wait for bond dissociation (*Figure 1—figure supplement 1C*). After that the target pipette would return to the original position to complete the cycle (*Figure 1—figure supplement 1Av–vii*). Each platelet was interrogated for a continuous time of 200 s to generate a force spectroscopy trace exemplified in *Figure 1C* before changing to a new pair of BFP bead and platelet. Lifetimes were measured from the instant when the force reached the desired level to the instant of bond dissociation (*Figure 1C*) (*Ju et al., 2015a*, *2013*). In the force-ramp mode, the force was loaded at different ramping rates (100, 1000 or 10,000 pN/s). The target was retracted continuously until bond rupture without holding at a constant force (*Figure 1—figure supplement 1D*).

## Protein unfolding analysis

Unfolding of GPIbα in the ramping phase was signified by a sudden force stagnation or drop (kink) as opposed to the linearly increasing force signals (*Figure 1E*). To determine the unfolding length, we derived a force vs. extension curve (*Figure 1E* inset) from the differential displacement between the BFP tracking system (probe position) and the piezoelectric actuator feedback system (target position) as previously described (*Chen et al., 2012*). The unfolding length was given by the sudden extension increase without force increase, the result of which was comparable to the differential contour length derived by fitting the prior- and post-unfolding force-extension curves with the worm-like chain (WLC) model (*Bustamante et al., 1994*) (*Figure 1—figure supplement 2*).

To reveal distinct populations, we used the nonparametric kernel density estimation to detect peaks in the data distribution of ensemble ramped unfolding lengths (*Freedman and Diaconis, 1981*) (*Figure 2—figure supplement 1A,B*) and used a reliable data-based bandwidth selection method (*Sheather and Jones, 1991*) to determine the optimal bin width as 5 nm for the data in *Figure 2D,F*. Using a 9.53 nm bin width determined by the Freedman-Diaconis formula and Freedman and Diaconis' heuristic rule (*Freedman and Diaconis, 1981*) for histogram analysis also revealed three peaks, although the valley separating the first two peaks consists of a single low fraction bin only (*Figure 2—figure supplement 1C*). The first two peaks in *Figure 2D* were suggested as MSD and LRRD unfolding, respectively, based on their favorable comparisons to the respective WM23 vs. platelet and A1WT vs. GC data in *Figure 2E*, which allowed only MSD or LRRD to unfoldrespectively. To test these hypotheses, we analyzed the molecular mechanics by sorting the unfolding forces and lengths into subgroups from the respective WM23 vs. platelet and A1WT vs. GC experiments, plotted the average unfolding force vs. length data, and fitted the data to the WLC model (*Zhang et al., 2009a*, *2015*). The best-fit curves were then served as standards to calibrate the average unfolding force vs. length data from the A1WT vs. platelet experiment (*Figure 2H,I*). The agreement between the data and the calibrated WLC curves rigorously verified the hypothetical identities of the first two peaks in *Figure 2D*.

Unfolding of GPIbα in the clamping phase was signified by a sudden force decrease (*Figure 1E*). The unfolding length was calculated from force change divided by BFP spring constant (*Figure 1F*), similar to the integrin extension length measurement from the previous distance-clamp analysis (*Chen et al., 2012*, *2016*). The duration from the beginning of the clamping phase to the beginning of unfolding was the time-to-unfold, $t_u$ (*Figure 1F*).

## Testing hypothesis for cooperative unfolding

The cooperativity between LRRD and MSD unfolding at a clamped force (e.g. 25 pN) was determined by testing the null hypothesis that the two domains unfolded independently. The frequencies of LRRD unfolding and MSD unfolding pulled by A1WT were calculated using data from *Figure 3—source data 1A* (take 25 pN for example):

|  | MSD = '+' | MSD = '−' | Row total |
|---|---|---|---|
| LRRD = '+' | 14 (9.7%) | 11 (7.6%) | 25 (17.2%) |
| LRRD = '−' | 25 (17.2%) | 95 (65.5%) | 120 (82.8%) |
| Column total | 39 (26.9%) | 106 (73.1%) | 145 (100%) |

The Pearson's $\chi^2$ test was used to test the null hypothesis (H$_0$) that LRRD unfolding and MSD unfolding are independent. The $\chi^2$ statistic was calculated as follow:

$$\chi^2 = \sum_{i=1}^{2} \sum_{j=1}^{2} \frac{(O_{ij} - E_{ij})^2}{E_{ij}} = 13.01$$

where $O_{ij}$ is the observed count and $E_{ij}$ is the expected count under the null hypothesis. The subscripts i and j denote LRRD and MSD respectively, whose values 1 and 2 denote with (+) and without (-) unfolding respectively. The system has $(2-1) \times (2-1) = 1$ degree of freedom. The small p-value ($3.09 \times 10^{-4}$ from the above $\chi^2$) requires that we reject the null hypothesis and accept the alternative hypothesis that LRRD and MSD unfolding are not independent. In other words, cooperativity exists between LRRD and MSD unfolding when GPIbα on platelets was pulled by A1WT. Similar statistical analyses were employed to assess cooperative unfolding between LRRD and MSD when GPIbα was pulled by A1WT, A1R1450E or AN51 at different clamped forces. The levels of significance were presented as $-\log10$ (p-values) (*Figure 3G,H*).

## Model for observing MSD clamped unfolding

The measured MSD time-to-unfold $t_u$ distributed as a single exponential decay: $p_u(t_u) = k_u e^{-k_u t_u}$, where $p_u$ is the probability density and $k_u$ is the unfolding rate of MSD under a clamping force. By fitting the semi-log plotted experimental distribution with a straight line (*Figure 4—figure supplement 1A*), the unfolding rate at 25 pN was evaluated from the negative slope or the reciprocal average time-to-unfold, $k_u = 1/<t_u> = 0.870 s^{-1}$. Modeling the force-dependent MSD unfolding rate (*Figure 4A*) by the Bell equation (*Bell, 1978*), we found the zero-force unfolding rate and the width of the energy barrier to be 0.26 s$^{-1}$ and 0.242 nm for pulling GPIbα via A1 on a live platelet. The first value is much larger and the second value is much smaller than the respective values previously obtained using an optical tweezer to measure ramped unfolding of MSD in purified GPIbα constructs (*Zhang et al., 2015*).

We previously reported that the A1–GPIbα bond lifetime $t_b$ distributed as a dual exponential decay with a fast- and a slow-dissociating off-rate (*Ju et al., 2013*). We also recently showed that unfolding of LRRD prolongs A1–GPIbα bond lifetime (*Ju et al., 2015b*). These results were also observed in this work, which comprise individual bond lifetime measurements that give rise to the averaged results in *Figure 4B,C*. Therefore, the probability densities for a A1–GPIbα bond, with and without prior LRRD unfolding in the ramping phase, to dissociate at time $t_b$ during the clamping phase are:

$$\mathrm{For\,LRRD-} : p_1(t_b) = w_{11}k_{11}e^{-k_{11}t_b} + w_{12}k_{12}e^{-k_{12}t_b} \tag{1}$$

$$\mathrm{For\,LRRD+} : p_2(t_b) = w_{21}k_{21}e^{-k_{21}t_b} + w_{22}k_{22}e^{-k_{22}t_b} \tag{2}$$

where $k_{ij}$ and $w_{ij}$ ($w_{i1} + w_{i2} = 1$) denote, respectively, off-rates and associated fractions of bonds under a clamped force, with the first subscript indicating without (*Equation 1*) or with (*Equation 2*) a prior LRRD unfolding event and the second subscript indicating the fast (*Equation 1*) or slow (*Equation 2*) dissociation pathway. By fitting the above model to the lifetime ensemble data, the parameters were calculated (*Figure 4—source data 1*).

Assuming that MSD unfolding and A1–GPIbα unbinding are independent events, the joint probability density for MSD unfolding at time $t_u$ and A1–GPIbα unbinding at time $t_b$ is $p(t_u, t_b) = p_u(t_u) \times p_i(t_b)$ where $i = 1, 2$ depending on whether LRRD unfolding occurs. This joint probability is depicted as a surface in *Figure 4D* and *Figure 4—figure supplement 1B*, using respective A1WT and A1R1450E data measured at 25 pN clamped force. The condition for observing MSD clamped unfolding is that the A1–GPIbα bond lifetime $t_b$ lasts longer than the time-to-unfold $t_u$. Thus, the probability of observing MSD unfolding in the clamping phase $P_{ui}$ is the volume under the probability density surface over the region $0 < t_u < t_b < \infty$, which is marked by the vertical red planes in *Figure 4D* and *Figure 4—figure supplement 1B*. For instance, in the absence of prior LRRD unfolding, the probability of observing MSD unfolding in the clamping phase of 25 pN is:

$$P_{\mathrm{u}1} = \int\limits_{0}^{+\infty} \left[ p_1(t_{\mathrm{b}}) * \int\limits_{0}^{t_{\mathrm{b}}} p_u(t_{\mathrm{u}}) dt_{\mathrm{u}} \right] dt_{\mathrm{b}} = \frac{w_{11}k_{\mathrm{u}}}{k_{\mathrm{u}} + k_{11}} + \frac{w_{12}k_{\mathrm{u}}}{k_{\mathrm{u}} + k_{12}} = 21.5\% \tag{3}$$

Similarly, in the presence of prior LRRD unfolding,

$$P_{\mathrm{u}2} = \frac{w_{21}k_{\mathrm{u}}}{k_{\mathrm{u}} + k_{21}} + \frac{w_{22}k_{\mathrm{u}}}{k_{\mathrm{u}} + k_{22}} = 46.2\% \tag{4}$$

The model was applied to predict the MSD clamped unfolding probabilities under different clamped forces pulled by A1WT and A1R1450E (*Figure 4E,F*). For A1R1450E, the ensemble MSD clamped unfolding events were no longer segregated into LRRD- and LRRD+ groups, because few MSD clamped unfolding events occurred following LRRD unfolding due to the reduced bond lifetime.

## Threshold and sensitivity-specificity analysis

We used the sensitivity-specificity analysis to solve the optimal threshold $t_0$ for pre-$Ca^{2+}$ longest lifetime $t_{max}$ separating $\alpha$ and $\beta$ $Ca^{2+}$ types. There are 4 possible outcomes (*Figure 5—figure supplement 2A*): a false positive (FP) happens when $t_{max} > t_0$ and a $\beta$-type $Ca^{2+}$ was observed; a false negative (FN) happens when $t_{max} \leq t_0$ and an $\alpha$-type $Ca^{2+}$ was observed; a true positive (TP) happens when and an $\alpha$-type $Ca^{2+}$ was observed, and a true negative (TN) happens when $t_{max} \leq t_0$ and a $\beta$-type $Ca^{2+}$ was observed. The sensitivity or true positive rate defines the fraction of true positive among all positive results, TP/(TP+FN), whereas the specificity or true negative rate defines the fraction of true negative among all negative results, TN/(TN+FP). The optimal threshold is solved by minimizing the total counts of false positive and false negative. To do that, a receiver operating characteristic (ROC) curve was created by plotting the TP rate (sensitivity) against the FP rate (1- specificity) at various $t_{max}$ values from which the optimal threshold $t_0$ that achieved the best sensitivity and specificity was identified (*Figure 5—figure supplement 2B*).

## Statistical analysis

Two-tailed Students' t-test was used to assess significance for group comparisons. Pearson correlation coefficient was used as a measure for linear dependency between two variables ($Ca^{2+}$ level and binding kinetics).

To determine errors in classification of different $Ca^{2+}$ types (null, $\beta$, $\alpha$) and in identification of unfolding (no unfolding, LRRD, MSD, MSD+LRRD), we assume that observed counts $(n_1, n_2, \ldots, n_{\mathrm{K}})$ follow a multinomial distribution with total counts $n = n_1 + n_2 + \ldots + n_{\mathrm{K}}$ and event probabilities $(p_1, p_2, \ldots, p_{\mathrm{K}})$. We then use the fraction of the $i$-th category, $n_i/n$, as an estimate for the $i$-th event probability $p_i$ and $\sqrt{(n_i/n)(1 - n_i/n)/n}$ as the associated standard error s.e.m.

## Acknowledgements

We thank MA Cruz for providing VWF-A1 proteins; MC Berndt and R Li for providing the WM23 antibody and personal communications; W Lam lab for the blood draw; SP Jackson lab for constructive comments and financial support to prepare the manuscript; J Lou for discussion on GPIbα structure and WLC model fitting; F Zhou for generating 3D probability density surface plots. B Liu for sharing the $Ca^{2+}$ analysis Matlab codes; W Chen, J Hong, Q Ji, and C Ge for fBFP instrumentation supports; J-F Dong, RP McEver, LV McIntire, R Andrews, H Lu and J Liao for helpful comments. This work was supported by NIH grant HL132019 (CZ), HL062350 (XD), HL080264 (XD) and HL125356 (XD), Diabetes Australia research grant G179720 (LJ), Sydney Medical School 2016 early-career researcher kickstart grant (LJ), and NSF grant DMS-1505256 (LX).

## Additional information

### Funding

| Funder | Grant reference number | Author |
| --- | --- | --- |
| Diabetes Australia | IRMA G179720 | Lining Ju |

| University of Sydney | 2016 Sydney Medical School ECR Kickstart Grant | Lining Ju |
| --- | --- | --- |
| National Science Foundation | DMS-1505256 | Lingzhou Xue |
| National Heart, Lung, and Blood Institute | HL062350 | Xiaoping Du |
| National Heart, Lung, and Blood Institute | HL080264 | Xiaoping Du |
| National Heart, Lung, and Blood Institute | HL125356 | Xiaoping Du |
| National Heart, Lung, and Blood Institute | HL132019 | Cheng Zhu |

The funders had no role in study design, data collection and interpretation, or the decision to submit the work for publication.

## Author contributions

LJ, Wrote and revised the paper, Performed research and analyzed data, Analysis and interpretation of data, Designed research, Contributed unpublished essential data or reagents; YC, Wrote and revised the paper, Performed research and analyzed data, Acquisition of data, Analysis and interpretation of data; LX, Performed statistical analysis, Analysis and interpretation of data, Drafting or revising the article; XD, Provided key reagents, Contributed unpublished essential data or reagents; CZ, Wrote and revised the paper, Designed research, Analysis and interpretation of data

## Author ORCIDs

Cheng Zhu, http://orcid.org/0000-0002-1718-565X

## Ethics

Human subjects: Human RBCs and platelets for BFP experiments were collected abiding a protocol (#H12354) approved by the Institute Review Broad of Georgia Institute of Technology. Informed consent was obtained from each blood donor.

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
