## [Decision Letter]

Thank you for submitting your article "Cooperative unfolding of distinctive mechanoreceptor domains transduces force into signals" for consideration by *eLife*. Your article has been reviewed by two peer reviewers, and the evaluation has been overseen by a Reviewing Editor and Richard Aldrich as the Senior Editor. The following individuals involved in review of your submission have agreed to reveal their identity: Fred Sachs (Reviewer #3).

The reviewers have discussed the reviews with one another and the Reviewing Editor has drafted this decision to help you prepare a revised submission.

Summary:

This paper uses an impressive combination of force and fluorescence measurements to investigate the role of tensile force on VWF-GPIbα induced Ca^2+^ signaling in platelets. First, using stand-alone BFP force measurements, the authors capture the unfolding of LRRD and MSD domains. Next, by combining BFP and fluorescence imaging, they monitor force induced Ca^2+^ fluxes in cells. Their results demonstrate that the unfolding of LRRD prolongs the lifetime of VWF-GPIbα binding and facilitates the unfolding of MSD domain which regulates the strength and type of Ca^2+^ release. This phenomenon is dependent on both the strength and duration of the force being applied. This paper has the potential to be very useful in understanding how cells transduce mechanical forces into biochemical signals. Several major concerns need to be addressed before the paper is considered further for publication. The manuscript is written in an unusually opaque way compared to other excellent manuscripts from the Zhu lab. The paper needs a rewrite to make it more accessible. Otherwise, its impact will be very limited.

Essential revisions:

1) In typical single molecule force measurements, the unfolding of a protein domain is characterized by an abrupt drop in force followed by a non-linear stretching of the unfolded polypeptide chain. The stretching of the polypeptide chain is then fitted to a Worm-Like-Chain (WLC) model to extract the contour length of the unfolded protein. However, in this manuscript, as shown in Figure 1, the authors use a 'kink' in the force measurement as an indication that the LRRD domain has unfolded; the characteristic WLC like stretching of the unfolded LRRD domain is not observed. Why is that? The authors need to fit the force vs. extension unfolding traces to a WLC model in order to convincingly claim that the LRRD domain is being unfolded (it would also be more accurate to calculate the contour length from fitting). Furthermore, do the authors have any independent evidence that the LRRD domain has unfolded in their experiments?

2) In Figure 2, the authors claim that the contour length histograms have three peaks. It is well recognized that multiple peaks in a histogram can merely be an artifact of an incorrectly chosen bin width. How were the bin widths in the histograms chosen? There are many well established statistical methods for calculating bin-widths and t the authors must use one of those methods to calculate bin sizes. For instance, when we eyeball the data presented in Figure 2 and calculate the bin width using the Freedman-Diaconis formula (a commonly used method for bin size estimation), we end up with a bin width of 15 pN. Using this bin width the data in Figure 2 will not have 3 peaks (in fact we suspect only one peak will be seen there).

3) We are confused by the force ramp unfolding measurements (Figure 3) showing the both the frequency of unfolding and the unfolding force increase with increasing force set-points (which the authors refer to as 'clamped force'). These appear to be force ramp measurements where the probe and the platelet are separated continuously and not a force clamp measurement. Why then should this data depend on the set point unless the different loading rates (separation velocity) were used for different set points? However the authors claim that the experiments were performed at the same loading rate. Furthermore, we are unable to reconcile the data in Figure 3 showing that no LRRD unfolding is observed when the force set-point is at 10 pN and the data in Figure 3 showing LRRD unfolding events when the set-point is 10 pN. The authors need to clear this up.

---

## [Author Response]

*Summary:*

This paper uses an impressive combination of force and fluorescence measurements to investigate the role of tensile force on VWF-GPIbα induced Ca^2+^ signaling in platelets. First, using stand-alone BFP force measurements, the authors capture the unfolding of LRRD and MSD domains. Next, by combining BFP and fluorescence imaging, they monitor force induced Ca^2+^ fluxes in cells. Their results demonstrate that the unfolding of LRRD prolongs the lifetime of VWF-GPIbα binding and facilitates the unfolding of MSD domain which regulates the strength and type of Ca^2+^ release. This phenomenon is dependent on both the strength and duration of the force being applied. This paper has the potential to be very useful in understanding how cells transduce mechanical forces into biochemical signals. Several major concerns need to be addressed before the paper is considered further for publication. The manuscript is written in an unusually opaque way compared to other excellent manuscripts from the Zhu lab. The paper needs a rewrite to make it more accessible. Otherwise, its impact will be very limited.

We thank the reviewers for the positive assessment of the significance of our work. The criticism regarding the manuscript presentation is well taken and we have carefully rewritten the manuscript. Many small changes were made throughout (rewording phrases, rearranging sentences and paragraphs, and re-structuring sections) to increase the accessibility to the general readers.

*Essential revisions:*

1) In typical single molecule force measurements, the unfolding of a protein domain is characterized by an abrupt drop in force followed by a non-linear stretching of the unfolded polypeptide chain. The stretching of the polypeptide chain is then fitted to a Worm-Like-Chain (WLC) model to extract the contour length of the unfolded protein. However, in this manuscript, as shown in Figure 1, the authors use a 'kink' in the force measurement as an indication that the LRRD domain has unfolded; the characteristic WLC like stretching of the unfolded LRRD domain is not observed. Why is that? The authors need to fit the force vs. extension unfolding traces to a WLC model in order to convincingly claim that the LRRD domain is being unfolded (it would also be more accurate to calculate the contour length from fitting).

We thank the reviewer for this excellent suggestion, which allows us to better relate our work to other studies of the single-molecule biophysics community.

We first fitted the WLC model to individual force-extension traces, as exemplified by the unfolding event shown in Figure 1. The WLC model fits both the force vs. extension data before and after unfolding (Bustamante et al., 1994) (Figure 1—figure supplement 2). The unfolding length calculated from the difference between the two best-fit contour lengths (318.6 – 279.3 = 39.3 nm) agrees well with that obtained by direct estimation (39 nm).

Due to the lower temporal resolution of our BFP (~1,500 fps) than the AFM (5-10 kHz scan rate) (Liang and Fernandez, 2009; Marshall et al., 2003) – a technique commonly used to study protein domain unfolding, the number of data points in the force-extension curve after unfolding is limited. As a result, the fitting parameters derived from individual pulling cycle may sometimes be variable. Therefore, we analyzed the ensemble data with the WLC model and fitted it to the averaged unfolding force vs. length data (Zhang et al., 2015; Zhang et al., 2009). MSD and LRRD have different contour lengths, and on platelet GPIbα they could unfold sequentially or concurrently. Before treating convoluted data from possible unfolding of either or both domains, we first analyzed data acquired from WM23 vs. platelet experiments, which contain only putative MSD unfolding events (Figure 2) and A1 vs. Glycocalicin (GC) experiments, which contain only putative LRRD unfolding events (Figure 2). From all ramped unfolding events, we sorted the unfolding lengths into 2-4 nm bins, plotted the averaged unfolding force vs. length curves, and fitted the WLC model to the data (Figure 2). It is evident that the WLC model fits both data sets well, yielding respective contour lengths of 25.99 ± 0.85 nm and 70.29 ± 3.56. Assuming a contour length of 4 Å per residue (Zhang et al., 2015), these values indicate that structural domains of 65 and 175 residues were respectively unfolded in the WM23 vs. platelet GPIbα and A1 vs. GC experiments, which well match the respective maximum lengths of unfolded MSD (Zhang et al., 2015) and LRRD (Ju et al., 2015) as previously discussed.

We next analyzed data from A1 vs. platelet experiments where unfolding of MSD, LRRD, or both were all possible. We have established a set of decision rules for segregating the observed unfolding events into these three groups based on simple “unfolding signatures” ([Supplementary-material SD1-data]). To provide a rigorous validation, we asked whether our criteria could stand the test of the WLC model. We grouped the unfolding lengths from the putative MSD and LRRD groups sorted according to our decision rules (the same data used in Figure 2) into 2-4 nm bins and overlaid the respective force vs. unfolding length plots on Figure 2, panels H and I to compare with data that just have been confirmed as MSD and LRRD unfolding and their WLC model fits. Remarkably, in both cases of MSD (Figure 2) and LRRD (Figure 2) the two force vs. unfolding length data (black circle and red triangle) are very well matched, validating our decision rules ([Supplementary-material SD1-data]). Note that to achieve good WLC model fitting, we acquired additional data from ramped unfolding events and we updated Figure 2 and Figure 3 using the additional data.

Figure 1—figure supplement 2 with Figure 2 and Figure 2. Validate GPIbα MSD and LRRD unfolding events using the WLC model. (Figure 1—figure supplement 2) Fitting of the WLC model to the force-extension traces (Figure 1 insert) before (blue) and after (red) the observed GPIbα ramped unfolding event. (Figure 2) The WLC model was fit (curves) to the unfolding force vs. length data (black circles, mean ± s.e.m. of 15-25 measurements per point) from the WM23 vs. platelet experiments where only MSD unfolding was possible (Figure 2) or A1 vs. GC experiments where only LRRD unfolding was possible (Figure 2), yielding a contour length of 25.99 ± 0.85nm or 70.29 ± 3.56nm, respectively. Overlying on the two panels are corresponding unfolding force vs. length data (red triangles, mean ± s.e.m. of 20-30 measurements per point) from A1 vs. platelet ramped experiments where unfolding of MSD, LRRD or both were all possible, but were segregated into putative MSD (Figure 2) and LRRD (Figure 2) unfolding groups based on our decision rules in [Supplementary-material SD1-data].

Furthermore, do the authors have any independent evidence that the LRRD domain has unfolded in their experiments?

In our previously published study, we have performed both steered molecular dynamics simulations (SMD) and the BFP pulling experiments to convincingly demonstrate LRRD unfolding (Ju et al., 2015). In SMD simulations, we stretched the GPIbα N-terminal domain (GPIbαN, Figure 8) with VWF-A1 and observed unfolding of leucine-rich repeats 2-4 of middle LRRD as force increased. In BFP experiments on the GC construct, we used antibody mapping to locate where unfolding happened. Both AN51 and HIP1 could induce GC unfolding events whereas SZ2 could not (Figure 8). This indicates that any pulling through a position lower than LRRD could not induce unfolding, confirming that unfolding occurred within the LRRD region. Interestingly, although the estimated contour length of LRRD is 70 nm, we rarely observed unfolding length beyond 56 nm with the GC construct, suggesting that not all leucine-rich repeats of the LRRD would be unfolded in every pulling of our BFP experiment.

Author response image 1.Domain organization and antibody epitopes on GPIbα.**DOI:**
http://dx.doi.org/10.7554/eLife.15447.024

2) In Figure 2, the authors claim that the contour length histograms have three peaks. It is well recognized that multiple peaks in a histogram can merely be an artifact of an incorrectly chosen bin width. How were the bin widths in the histograms chosen? There are many well established statistical methods for calculating bin-widths and t the authors must use one of those methods to calculate bin sizes. For instance, when we eyeball the data presented in Figure 2 and calculate the bin width using the Freedman-Diaconis formula (a commonly used method for bin size estimation), we end up with a bin width of 15 pN. Using this bin width the data in Figure 2 will not have 3 peaks (in fact we suspect only one peak will be seen there).

We thank the reviewers for the excellent suggestion that we choose the bin width using well-established statistical methods, which gives us an opportunity to validate the appropriateness of the bin width we used.

The Freedman-Diaconis formula is commonly used for estimating the bin width of histograms. According to Freedman and Diaconis (1981), the optimal bin width is proportional to the inverse of the cube root of sample size *N* (i.e.,) when is sufficiently large. However, the *k* valuein this formula is unknown, making it challenging in practice. Freedman and Diaconis suggested the heuristic rule to choose *k* as twice the interquartile range of the data. Using this rule we calculated *k =* 52.88 and a bin width of 9.53 nm from our data of *N* = 171 (Figure 2). Note that the histogram plotted using this bin width still reveals three peaks, although the valley separating the first two peaks consists of a single low fraction bin only (Figure 2—figure supplement 1).

We note that the purpose of analyzing unfolding length distribution is to correctly identify heterogeneous subpopulations or multiple peaks in the probability density estimates. It is well-known that kernel density estimates are the best nonparametric estimator of the probability density of the underlying population (Wahba, 1975), and they provide a better approximation than histograms given the limited sample size (Scott, 1979). With this rationale, we first used the nonparametric kernel density estimation method to choose the bin width and detect peaks in the data distribution, without imposing any parametric assumptions (Freedman and Diaconis, 1981). Specifically, we used the well-established reliable data-based bandwidth selection method (Sheather and Jones, 1991) to determine the optimal bin width to be 5.12 nm. As shown in Figure 2—figure supplement 1, using Gaussian kernels and Epanechnikov kernels both result in kernel density estimates that resemble the Figure 2 histogram and identify three peaks in the ramped unfolding length distribution. This analysis has validated our choice of 5 nm as the bin width and confirm the presence of three peaks in the histograms. The discussion has been incorporated in the Method.

Figure 2—figure supplement 1. Statistical analysis on ramped unfolding length distribution, related to Figure 2. (A) Histogram analysis on ramped unfolding length distribution using the bin width 9.53 nm obtained using the Freedman-Diaconis formula and Freedman and Diaconis’ heuristic rule. (B,C) Nonparametric kernel density estimation using the software R with the given kernel ("gaussian" or "epanechnikov") and bandwidth selected by Sheather and Jones (1991). Both estimates identify three populations in the distribution of the ramped unfolding length data from A1 vs. platelet GPIbα experiments.

3) We are confused by the force ramp unfolding measurements (Figure 3) showing the both the frequency of unfolding and the unfolding force increase with increasing force set-points (which the authors refer to as 'clamped force'). These appear to be force ramp measurements where the probe and the platelet are separated continuously and not a force clamp measurement. Why then should this data depend on the set point unless the different loading rates (separation velocity) were used for different set points? However the authors claim that the experiments were performed at the same loading rate. Furthermore, we are unable to reconcile the data in Figure 3 showing that no LRRD unfolding is observed when the force set-point is at 10 pN and the data in Figure 3 showing LRRD unfolding events when the set-point is 10 pN. The authors need to clear this up.

We apologize for the confusing data here. The experiments were performed in force-clamp assays (See Method) but the unfolding events in Figure 3 occurred during the ramping phase before the force arrived at the clamped level. After checking, we found that the original Figure 3 was mistakenly labeled. The LRRD unfolding events should have been collected with experiments pulling GPIbα on platelets rather than its purified extracellular portion (GC). Here we corrected the label in Figure 3 and replaced the Figure 3 with the platelet GPIbα data. We did not observe any LRRD unfolding on live platelets at 10 pN clamped force, but did observe (although extremely infrequent, ~2%) LRRD unfolding using purified GC in Figure 3—figure supplement 1.